# CONSISTENCY GUARANTEED CAUSAL GRAPH RECOVERY WITH LARGE LANGUAGE MODELS

## ABSTRACT

Causal graph recovery traditionally relies on statistical estimation of observable variables or individual knowledge, which suffer from data collection biases and knowledge limitations of individuals. Leveraging the broad knowledge in scientific corpus, we propose a novel method for causal graph recovery to deduce causal relationships with the large language models (LLMs) as a knowledge extractor. Our method extracts associational relationships among variables and further eliminates the inconsistent relationship to recover a causal graph using the constraint-based causal discovery methods. Comparing to other LLM-based methods that directly instruct LLMs to do highly complex causal reasoning, our method shows advantages on causal graph quality on benchmark datasets. More importantly, as causal graphs may evolve when new research results emerge, our method shows sensitivity to new evidence in the literature and can provide useful information to update causal graphs accordingly.

## 1 INTRODUCTION

Estimating causal effect between variables from observational data is a fundamental problem to many domains including medical science (Höfler, 2005), social science (Angrist et al., 1996), and economics (Imbens & Rubin, 2015; Yao et al., 2021). It enables reliable decision-making from complex data with entangled associations.

While it is usually expensive and infeasible to investigate causal effects using randomized experiments, researchers employ causal inference (Pearl, 2010) to estimate causal effects from observational data. There are two main frameworks for causal inference: the potential outcome framework (Rubin, 1974) and the structural causal model (SCM) (Pearl, 1995). Priori causal structures, usually represented as Directed Graphical Causal Models (DGCMs) (Pearl, 2000; Spirtes et al., 2001), are often used to represent and analyze the causal relationships. These causal graphs help disentangle the complex interdependencies and facilitate the analysis of causal effects. Recovering causal graphs often relies on experts' knowledge or statistical estimation on observational data (Spirtes & Glymour, 1991). Causal Discovery (CD) algorithms (Spirtes & Glymour, 1991) are the main statistical estimation-based methods that use conditional independence tests to assess conditional associational relationships (called associational reasoning) for inferring causal connections (Spirtes et al., 2001; Chickering, 2002; Shimizu et al., 2006; Sanchez-Romero et al., 2018).

Consequently, the reliability of these algorithms is affected by the quality of data, which can be compromised by issues such as data collection bias (Zhang et al., 2017; Bareinboim et al., 2014; Bhattacharya et al., 2021) (See Example 1 in Appendix A.1). Additionally, CD algorithms often assume certain distribution, such as Gaussian about data, which may fail to accurately reflect the complexity of real-world scenarios.

To overcome these limitations, Large Language Models (LLMs) (Zhao et al., 2023) have been employed for causal graph recovery. While few studies have explored hybrid solutions that use LLMs to refine the results of statistical estimation-based methods (Vashishtha et al., 2023; Ban et al., 2023), most work relies solely on LLMs to output causal graphs. Among these, one way is to directly ask LLMs to infer the conditional associational relationships (CARs) between each pair of factors in the graph to build the causal graph (Choi et al., 2022; Long et al., 2022; Kıcıman et al., 2023), relying only on the LLM's background knowledge. However, it is questionable whether LLMs possess sufficient domain-specific knowledge or causal reasoning capabilities to perform this task

effectively (Kandpal et al., 2023; Zečević et al., 2023). An alternative methodology is to inject the causal graph knowledge into LLMs (Kandpal et al., 2023; Zečević et al., 2023). However, studies such as (Cohrs et al., 2023) have reported low LLM performance in recognizing CARs. Experiments conducted by (Jin et al., 2023b) show that fine-tuning LLMs on synthetic CAR datasets can improve performance on trained tasks. However, they also demonstrate that the LLM is merely rote learning the CARs from the synthetic data rather than learning how to reason about CARs, as evidenced by significant performance drops when variable names are changed.

We propose the LLM Assisted Causal Recovery (LACR) method to address the challenges faced by current causal graph recovery approaches. Instead of relying on LLMs' ability to perform complex causal reasoning, LACR capitalizes on their strength in understanding and extracting information from vast amounts of scientific literature. By doing so, we leverage the LLMs' ability to interpret complex associational and causal insights hidden in a large scientific corpus, rather than relying solely on their reasoning capabilities.

LACR retrieves relevant knowledge from a comprehensive scientific corpus that contains valuable information about the relationships between variables. The LLM is used to infer how each document supports or refutes the conditional associational relationship (CAR) between two factors, extracting CAR estimations based on the evidence provided by the retrieved literature. This retrieval-based strategy allows us to build a rich dataset of CAR estimations that are grounded in scientific knowledge and experimental data, which helps us to overcome the data collection bias problem.

By aggregating the CAR estimations returned by the LLMs, LACR recovers the causal graph through a constraint-based causal discovery algorithm. The aggregation process is not arbitrary; instead, it is formalized as a collective decision-making problem, ensuring that the most consistent CAR estimations are retained while maintaining an acyclic structure for the graph. We demonstrate that this problem is NP-hard and provide approximation algorithms to address it effectively.

We validate the effectiveness of LACR through extensive experiments on two well-known real-world causal graphs. Our results show that LACR not only recovers accurate causal graphs but also identifies biases in validation datasets commonly used in the causal discovery community. This highlights the potential of LACR to recover causal graphs that are better aligned with the latest domain knowledge, suggesting avenues for improving current validation practices in causal discovery.

## 2 BACKGROUND

We first introduce the preliminaries of the *directed graphical causal models* and the *causal graph recovery* problem.

### 2.1 DIRECTED GRAPHICAL CAUSAL MODELS (DGCMs)

A DGCM is a tuple $M = \langle G, P \rangle$, in which, $G = \langle V, E \rangle$ is a Directed Acyclic Graph (DAG), also known as a *causal graph*. The set of nodes $V = \{v_1, \cdots, v_n\}$ represents random variables (with $|V| = n$), and $E \subseteq \{(v_i, v_j) \mid v_i, v_j \in V, v_i \neq v_j\}$ are directed edges, also called *causal edges*, that encode *causal relationships*. Let $\bar{G} = \langle V, \bar{E} \rangle$ be the *skeleton* of DAG $G$, where each $(v_i, v_j) \in \bar{E}$ is an undirected edge, and it indicates that one of $(v_i, v_j)$ and $(v_j, v_i)$ is in $E$. Given a variable set $V$, we denote the set of all DAGs and all skeletons by $\mathcal{G}$ and $\bar{\mathcal{G}}$, respectively. In $G$, let a sequence of distinct nodes $\ell = (v_{j_1}, v_{j_2}, \cdots, v_{j_m})$ denote a *path*, such that for each $i \in \{1, 2, \cdots, m-1\}$, either $(v_{j_{i+1}}, v_{j_i})$ or $(v_{j_i}, v_{j_{i+1}}) \in E$. A path is a *causal path* from $v_{j_1}$ to $v_{j_m}$ if for each $i \in \{1, 2, \cdots, m-1\}, (v_{j_i}, v_{j_{i+1}}) \in E$. $P$ is a *joint probability distribution* of all variables in $V$. Note that in our method, we allow the existence of exogenous variables, i.e., variables not contained in $V$ may mediate the causal relationships between variables in $V$.

### 2.2 CONSTRAINTS OF CAUSAL GRAPHS

A causal graph is subject to a series of constraints on variables' *conditional associational relationships* (CARs). Especially, the causal edges specify the causal relationships between variables. $(v_i, v_j)$ represents that $v_i$ is a *direct cause* of $v_j$, i.e., when holding the other variables constant, varying the value of $v_i$ triggers a corresponding change in the value of $v_j$, but not vice versa. This causal relationship thus entails the associational relationship between the variables, i.e., their marginal

probability distributions $P(v_i)$ and $P(v_j)$ are associated (or correlated), which does not have the direction attribute. Note that two variables can be associated even though they do not have a direct causal relationship. Typical examples are that two variables linked by a causal path, and two variables pointed to by two causal paths that have the same starting node (which is usually called a covariate). The precise constraints follow the well known Causal Markov Assumption.

**Assumption 1** (Causal Markov Assumption). *In any causal graph, each variable is independent of its non-descendants conditioned on its parents in the causal graph, i.e., $v \perp non\_desc(v) \mid parent(v)$.*

The structure of a causal graph implies graphical constraints called *d-separation* (Pearl, 2000) that specify a conditional associational relationship between variables. In the rest of this paper, for any given variable pair $v_i, v_j \in V$, we constantly use $V'$ to denote an arbitrary subset of $V \setminus \{v_i, v_j\}$, unless otherwise specified. If $V'$ d-separates $v_i$ and $v_j$, then the joint probability distribution $P$ encodes that the two variables are independent conditioned on $V'$. We say the association between $v_i$ and $v_j$ is blocked by $V'$, and if $v_i$ and $v_j$ cannot be d-separated, their association is unblockable.

Assumption 1 is a necessary condition for the encoding of the associaitonal relationship constraints in $P$. On the other hand, the following *faithfulness assumption* is a sufficient condition that $P$ encodes such constraints.

**Assumption 2** (Causal Faithfulness Assumption). *A joint distribution $P$ does not encode additional conditional associational relationships other than those consistent with $G$'s d-separation information. We call such $P$ is faithful to $G$.*

We now formally define the constraints of distribution $P$ that is faithful to causal graph $G$. Such constraints are typically used in constraint-based causal recovery algorithms, such as the PC algorithm and the FCI algorithm. The principle of the constraints is that, a causal edge exists between a pair of variables if and only if this variable pair cannot be d-separated. Note that we say a variable pair is d-separated by an empty set if their marginal distributions are independent from each other.

Let $\alpha(ij \mid V') \in \{0, 1\}$ be the conditional associational relationship between variables $v_i, v_j \in V$ conditioned on variable set $V'$. $\alpha(ij \mid V') = 0$ denotes that $v_i$ and $v_j$ are independent conditioned on $V'$ according to $P$, and $\alpha(ij \mid V') = 1$ denotes associated. We write $\alpha(ij)$ when $V' = \emptyset$.

**Definition 1** (Constraints of causal graphs). *With Assumptions 1 and 2 , we have that for $v_i, v_j \in V$:* **1.** $V'$ *d-separates* $v_i$ *and* $v_j$ $\implies$ $\alpha(ij \mid V') = 0$; **2.** $\alpha(ij) = 0$ *or* $\exists V'$ *s.t.* $\alpha(ij \mid V') = 0$ $\implies$ $(v_i, v_j) \notin \bar{E}$; **3.** $\nexists V'$ *s.t.* $\alpha(ij \mid V') = 0$ $\implies$ $(v_i, v_j) \in \bar{E}$.

## 3 METHODOLOGY

In this section, we introduce our *large language model assisted causality recovery* (LACR) method, which runs in two phases: the causal edge existence verification, and the edge orientation. LACR employs LLMs to extract the CARs from relevant scientific literature, and recovers the causal graph using the constraint-based causal discovery principles. Specifically, LACR extracts CARs of variables from previous data analysis in relevant scientific literature, to investigate whether each variable pair can be d-separated. Then, it uses such extracted CARs to recover the causal graph based on the constraints of the causal graph (Definition 1).

Notably, in constraint-based causal discovery algorithms, if the dataset satisfies the faithfulness assumption, the obtained CARs do not conflict against each other. However, this may fail extracted CAR estimations from scientific literature, as CAR conflicts may arise due to the analysis noise introduced by previous scientific research and the noise introduced in the extraction process.

### 3.1 INCONSISTENT ASSOCIATIONS

Two types of CAR inconsistency may occur in our setting, namely the *causal existence inconsistency* and the *d-separation inconsistency*.
**Causal existence inconsistency** specifies the situations where for a specific pair of variables $v_i$ and $v_j$, part of the extracted CARs indicate that $\nexists V' \subseteq V \setminus \{v_i, v_j\}$ s.t. $\alpha(ij \mid V') = 0$, however, the other part indicate that $\exists V' \subseteq V \setminus \{v_i, v_j\}$ s.t. $\alpha(ij \mid V') = 0$. Note that it is possible that $V' = \emptyset$. On the other hand, **d-separation inconsistency** denotes the following conflict. For a variable pair

$v_i, v_j \in V$, we call $V' \in V \setminus \{v_i, v_j\}$ a *minimal* d-separation set if $\alpha(ij) = 1$, $\alpha(ij \mid V') = 0$, and $\nexists V'' \subset V'$ such that $\alpha(ij \mid V'') = 0$. Then, we first have the following lemma.

**Lemma 1.** *Let $V'$ be a minimal d-separation set of $v_i$ and $v_j$. Then, for each variable $v' \in V'$, $v'$ and $v_i$ are associated, and so do $v'$ and $v_j$.*

Note that all full proofs can be found in Appendix C. If a dataset is faithful to the underlying causal graph, Lemma 1 is satisfied. However, this cannot be guaranteed in our setting.

Apparently, if d-separation inconsistency can be avoided, it is straightforward to deal with the causal existence inconsistency problem by checking the extracted CARs for each variable pair separately. However, the process tackling the d-separation inconsistency needs involving the CARs of other variable pairs, which considerably enhance the computational complexity.

### 3.2 LACR 1: CAUSAL EDGE EXISTENCE VERIFICATION

Now, we are ready to introduce the first phase of LACR. In LACR 1, given a set of variables $V$, for each pair of variables $v_i, v_j \in V$, we first retrieve a fixed number of the most relevant scientific papers. Then, for each paper, we use the LLMs to extract the corresponding estimated CAR information, which we call a CAR estimation piece, as follows:

1. are $v_i$ and $v_j$ associated, i.e., the value of $\hat{\alpha}(ij)$?
2. If $\hat{\alpha}(ij) = 1$, does the paper indicate $v_i$ and $v_j$ can be d-separated by a variable set? That is, does it hold that $\exists V' \subseteq V \setminus \{v_i, v_j\}$ s.t. $\hat{\alpha}(ij \mid V') = 0$?
3. If $\exists V' \subseteq V \setminus \{v_i, v_j\}$ s.t. $\hat{\alpha}(ij \mid V') = 0$, find a minimal d-separation set.

Note that both causal existence inconsistency and d-separation inconsistency potentially occur in the above extracted CARs. We therefore define and solve an optimization problem where we delete the least number of CAR estimation pieces to eliminate both types of inconsistency. Finally, we recover the skeleton of the causal graph following Definition 1.

#### 3.2.1 CAR EXTRACTION

We now introduce our strategy for CAR estimation piece extraction. We aim at designing a CAR estimation piece extraction workflow with the least task specific prompt, so to maintain the generalization ability of the workflow. In the workflow, we first retrieve a fixed number of the most relevant scientific papers from scientific literature databases, and we query the LLMs to extract desirable CAR estimation from each paper, and to respond in a structured format.

**Scientific document retrieval** Given the variable set $V$, for each variable pair $v_i, v_j \in V$, we retrieve relevant scientific papers, called scientific documents, from databases. We rank the retrieved scientific documents based on a matching function, e.g., a key word matching function or a semantic matching function, between each document and the paper searching query "$v_i$ and $v_j$", for each variable pair, and store the first $k$ documents in set $\mathbf{DOC}_{ij}$. Let $\mathbf{DOC} = \{\mathbf{DOC}_{ij}\}_{v_i, v_j \in V}$.

**CAR extraction prompt strategy** We design a series of prompts to query the LLMs to extract CAR estimation pieces from retrieved scientific documents, including a task background reminding prompt, and two CAR context prompts. We first prepare the extraction process by making the LLMs clarify the correct meaning of each variable with extra input of the domain names from which the variables are from, e.g., biology, medical science, and social science. Specifically, we simply query the LLMs by prompt "Clarify the meaning of each factor in $V$, which are from the domains of ...". Then, we let the LLMs to understand the first CAR context, the association context, which instructs the intuition and frequently used descriptions of whether $v_i$ and $v_j$ are associated or not, and to extract $\hat{\alpha}(ij)$. Upon extracted $\hat{\alpha}(ij) = 1$, the process moves to query the LLMs to understand the second CAR context, the association type context, which provides the intuition and frequently used descriptions of whether $v_i$ and $v_j$ can be d-separated, and to extract whether there exists $V' \subseteq V \setminus \{v_i, v_j\}$ s.t. $\hat{\alpha}(ij \mid V') = 0$.

**CAR extraction** Based on the above key prompts, we use Algorithm 1 to extract a CAR estimation piece from each retrieved document if it contains such analyzing result. Intuitively, for each document or LLM's background knowledge, i.e., KB on Line 3, we query LLM to extract if the KB

---

**Algorithm 1** CAR extraction

---

1: **Initialization:** $V, \mathbf{DOC}, \mathbf{S} = \{S_{ij} = \emptyset \mid v_i, v_j \in V\}, \mathbf{DS} = \{DS_{ij} = [\emptyset, \emptyset] \mid v_i, v_j \in V\}$
2: **for** $v_i, v_j \in V$ **do**
3:     **for** `KB` $\in \mathbf{DOC}_{ij} \cup \{\texttt{BG}\}$ **do**
4:         **if** $\hat{\alpha}_{\texttt{KB}}(ij) = 0$ **then**
5:             $S_{ij} = S_{ij} \cup \{\texttt{all}\}$
6:         **else if** $\hat{\alpha}_{\texttt{KB}}(ij) = 1$ **then**
7:             **if** $\exists V' \subseteq V \setminus \{v_i, v_j\}$ s.t. $\hat{\alpha}_{\texttt{KB}}(ij \mid V') = 0$ **then**
8:                 $S_{ij} = S_{ij} \cup \texttt{min}(V')$
9:                 $V^i = \emptyset, V^j = \emptyset$
10:               **for** $v_k \in V'$ **do**
11:                   **if** $\hat{\alpha}_{\texttt{KB}}(ik) = 1$ and $\nexists V'' \subseteq V \setminus \{v_i, v_k\}, \hat{\alpha}_{\texttt{KB}}(ik \mid V'') = 0$ **then**
12:                       $V^i = V^i \cup \{v_k\}$
13:                   **else if** $\hat{\alpha}_{\texttt{KB}}(jk) = 1$ and $\nexists V'' \subseteq V \setminus \{v_j, v_k\}, \hat{\alpha}_{\texttt{KB}}(jk \mid V'') = 0$ **then**
14:                       $V^j = V^j \cup \{v_k\}$
15:                 $DS_{ij}$.append($[V^i, V^j]$)
16:             **else if** $\nexists V' \subseteq V \setminus \{v_i, v_j\}$ s.t. $\hat{\alpha}_{\texttt{KB}}(ij \mid V') = 0$ **then**
17:                 $S_{ij} = S_{ij} \cup \{\texttt{none}\}$
18:         continue
19:     continue
20: **Return:** $\mathbf{S}, \mathbf{DS}$

---

indicates association or non-association between the variable pair. If the `KB` indicates association, LLM further investigates whether the association can be blocked or not (Lines 4-6), and we instruct LLM to return the corresponding d-separation set if the association can be blocked (Lines 7-17).

In Algorithm 1, we initiate the algorithm (Line 1) with two empty sets for each pair of distinct variables, and we use the d-separation collection $\mathbf{S}$ to record all estimated d-separation sets for each variable pair, and use $\mathbf{DS}$ to record subsets of each d-separation set, each element in which has an unblockable association with $v_i$ ($DS_{ij}[0]$) and an unblockable association with $v_j$ ($DS_{ij}[1]$), respectively. Then, we query the LLMs to extract CAR estimated piece from each retrieved document for $v_i$ and $v_j$, denoted as $\mathbf{DOC}_{ij}$, as well as the LLMs' background knowledge, denoted as `BG` from Lines 3 to 21. It is possible that the retrieved document or `BG` does not contain required information, where the LLMs return unknown, and we skip the document or the `BG` (Lines 4-5, and 20-21). We first ask the LLMs to extract the information whether $v_i$ and $v_j$ are associated. If the LLMs specify that the variable pair are independent, i.e., $\hat{\alpha}_{\texttt{KB}}(ij) = 0$, we record a d-separation set as `all` in $S_{ij}$, indicating that $v_i$ and $v_j$ can always be d-separated (Lines 6-7). Otherwise, we let the LLMs to further extract whether there exists a variable set $V'$ that d-separates $v_i$ and $v_j$. If the answer is positive, we ask the LLMs to return a minimal d-separation set of $V'$, i.e., $\texttt{min}(V')$, and record it in $S_{ij}$ (Lines 9-10). Next, we query the LLMs to check if each element in $V'$ has an unblockable association with $v_i$ or $v_j$, recording the element in $DS_{ij}[0]$ or $DS_{ij}[1]$, respectively, if it does. If no separation set is found, we record `none` in $S_{ij}$, indicating $v_i$ and $v_j$ cannot be d-separated. Algorithm 1 finally returns a d-separation collection $\mathbf{S}$ and a corresponding $\mathbf{DS}$. As follows, we show that the output of Algorithm 1 rigidly maps to the constraints of a causal graph (Definition 1).

**Proposition 1.** *For each variable pair $v_i$ and $v_j$, the mapping from their CAR space to the space of the returned d-separation set is a surjection, and the mapping from the space of the d-separation set to the space of causal edge existence, i.e., whether $(v_i, v_j) \in \bar{E}$ or not, is also a surjection.*

### 3.2.2 CONSTRAINT-BASED CAUSAL EDGE EXISTENCE VERIFICATION

Now, we recover the causal graph skeleton by the d-separation collection $\mathbf{S}$ returned by Algorithm 1. Since the inconsistency issues occur, we formulate the causal edge existence verification process as a collective decision making problem through an *approval voting* instance (Brandt et al., 2016). Each d-separation set $s \in \{\cup S_{ij}\}_{v_i, v_j \in V}$ for all $v_i, v_j \in V$ casts a vote over all possible skeletons in $\bar{\mathcal{G}}$. For each d-separation set $s \in S_{ij}$, its vote $b_s \in \{0, 1\}^{2^{n(n-1)/2}}$ is an approval vote, that assigns score 1 to a skeleton if $s$ approves it, otherwise assigns score 0 to the skeleton. $s$ approves a skeleton $\bar{G} = \langle V, \bar{E} \rangle$ if

1. $s = \texttt{none}$ and $(v_i, v_j) \in \bar{E}$; or
2. $s = \texttt{all}$ and $(v_i, v_j) \notin \bar{E}$; or
3. $s = V'$, and $(v_i, v_j) \notin \bar{E}$ and for all $v \in V'$, both of $(v_i, v) \in \bar{E}$ and $(v_j, v) \in \bar{E}$ hold.

**Objective** We aim at selecting the skeleton that obtains the highest score. When there is a tie, we break the tie by selecting the skeleton with fewer edges.

By utilizing an approval voting instance to select the skeleton, we eliminate the inconsistency issue by discarding the least number of extracted CAR estimation pieces. Note that the approval voting result is slightly biased towards not retaining an edge due to the tie breaker, because in cases with high noise or lack of extracted estimations (e.g., when tie happens), it tends to be that no unblockable association exists.

Given a d-separation collection $\mathbf{S}$, the problem of selecting the skeleton with the most approvals can be reduced to the following problem in polynomial time. The d-separation collection $\mathbf{S}$ may give rise to inconsistency issues, and therefore, we aim at maximizing the number of adopted d-separation sets in $\mathbf{S}$ subject to the following constraints.

**Definition 2** (causal consistent constraints). *Given a d-separation collection $\mathbf{S}$, we adopt a subset of $\mathbf{S}$ that is causal consistent if the subset satisfies the following constraints.*
*(1) for each variable pair $v_i$ and $v_j$, only one of two d-separation set types can be adopted: (1) $s = \texttt{none}$; or (2) $s = \texttt{all}$ or $s = V'$ and $|V'| > 0$. The two d-separation set types correspond to whether (1) $(v_i, v_j) \in \bar{E}$; or (2) $(v_i, v_j) \notin \bar{E}$, and therefore this constraint eliminates the causal existence inconsistency.*
*(2) Let $s \in S_{ij}$ for an arbitrary variable pair $v_i$ and $v_j$ such that $s = V'$. Then $s$ cannot be adopted concurrently with $s'$ which satisfies: (i) $s' \in S_{ik}$, and $v_k \in V'$ and $s' = \texttt{all}$, or $v_k \in V^i$ and $s' \neq \{\texttt{none}\}$; (ii) $s' \in S_{jk}$, and $v_k \in V'$ and $s' = \{\texttt{all}\}$, or $v_k \in V^j$ and $s' \neq \texttt{none}$; (iii) for any $v_k \in DS_{ij}[0]$ (resp. $v_k \in DS_{ij}[1]$), $s' \in S_{ik}$ (resp. $s' \in S_{jk}$) such that $s' \neq \{\texttt{none}\}$ This constraint eliminates the d-separation inconsistency.*

Then, we can define the optimization problem, namely the maximizing consistency (MAXCON) problem as follows.

**Definition 3** (MAXCON). *Given a set of variables $V$ and a d-separation collection $\mathbf{S}$, for each variable pair $v_i, v_j \in V$, let $\delta(s) = 1$ denote that $s \in S_{ij}$ is adopted, otherwise $\delta(s) = 0$, and let $\boldsymbol{\delta} = \{\delta(s) \mid s \in S_{ij}, \forall v_i, v_j \in V\}$. Then, the MAXCON aims at maximizing the adopted d-separation sets subject to the causal consistent constraints, i.e.,*

$$\underset{\boldsymbol{\delta}}{argmax} \sum_{\delta(s) \in \boldsymbol{\delta}} \delta(s) \tag{1}$$

$$s.t. \; \{s \in S_{ij} \mid v_i, v_j \in V, \delta(s) = 1\}, \; and \; causal \; consistent \; constraints \tag{2}$$

**Lemma 2.** *Given the solution of the MAXCON, it costs $\mathcal{O}(n^2)$ to compute the skeleton with the most approvals, where $n$ is the number of variables in $V$.*

**Theorem 1.** *The MAXCON problem is NP-hard.*

A nontrivial challenge is that MAXCON problem is NP-hard, as we shown in Appendix C.4. Therefore, we propose Algorithm 2, Inconsistency-Free MAXCON Algorithm, for the MAXCON problem, which is initiated with a conflict graph $CG = \langle CS, CE \rangle$ (please refer to Appendix C.5). Each node $s \in CS = \{s \in S_{ij} \mid v_i, v_j \in V\}$ in the conflict graph is a d-separation set in $\mathbf{S}$. A pair of nodes are connected in $CG$ is they cannot be adopted concurrently according to Definition 2.

**Theorem 2.** *Algorithm 2 has an approximation ratio of $\frac{1}{\Delta+1}$, where $\Delta$ is the maximum degree of the conflict graph $G$. That is, the size of the adopted votes produced by the algorithm satisfies $|S| \geq \frac{1}{\Delta+1}|\text{OPT}|$, where $\text{OPT}$ is the size of the maximum adopted votes without conflict.*

### 3.3 LACR 2: ORIENTATION

We further infer the orientation of edges based on the recovered skeleton. Similar to the previous step, we leverage a voting mechanism to decide the orientation of each edge in the skeleton, using the same set of scientific documents for each variable pair. However, since each edge is inferred

---

**Algorithm 2** Inconsistency-Free MAXCON Algorithm

---

1: **Input:** Conflict graph $CG = \langle CS, CE \rangle$
2: **Output:** D-separation collection **S**
3: Initialize an empty set $\mathbf{S} \leftarrow \emptyset$
4: **while** $V$ is not empty **do**
5:   Let $s$ be a node in $CS$ with the minimum degree in $CG$
6:   $\mathbf{S} \leftarrow \mathbf{S} \cup \{s\}$
7:   Remove $s$ and all its neighbors from $CG$
8: **return S**

---

individually, there may be inconsistencies in the orientation collection **D**, such as directional inconsistency or cyclic inconsistency. A straightforward approach is to order all edges by weight and process each edge from the highest to the lowest weight, orienting it based on the majority of orientation estimations. If this creates a cycle, we reverse the direction, and if a cycle still forms, the edge is removed. However, this method risks cascade failures, where an early misorientation of a high-weight edge could negatively affect the remaining orientations.

This problem is NP-hard, as it can be reduced from the Feedback Arc Set (FAS) problem, which aims to minimize the number of edges removed to make a directed graph acyclic. To address this, we propose Algorithm 3 (detailed in the Appendix), an approximation solution that selects a directed acyclic graph (DAG) with the maximal subset of consistent orientation estimations from **D**, while eliminating both directional and cyclic inconsistencies. Due to page limitations, we only provide an outline here. For the problem definition, algorithm, and all proofs, please refer to the Appendix D.2.

## 4 EXPERIMENTS

In this section, we provide experimental results on two practical benchmark datasets. *Most importantly*, we show some information is out-of-date in these datasets and how our results reflect recent scientific evidence associated with the datasets, which indicates the need of adjustment to the "ground truth" causal graph, and we validate LACR against the benchmark causal graphs that factor in the new evidence.

### 4.1 EXPERIMENT DATA

**Validation datasets.** We validate our method on the two largest small-scale networks, namely ASIA and SACHS, in the bnlearn package (Scutari et al., 2019). Both datasets have reported causal graphs (see Appendix E) based on real-world data. It is worth noting that, we only limit the selection of validation datasets to real-world datasets because LACR uses a real-world knowledge base.

**ASIA (lau, 1988).** The ASIA dataset has 8 nodes (from domains of medical, biology, and social science) and 8 edges, revealing the potential reasons and symptoms of lung diseases.
**SACHS (Sachs et al., 2005).** The SACHS dataset has 11 nodes (from the medical and biological domains) and 16 edges. It uncovers the interaction among proteins related to several human diseases.

### 4.2 EXPERIMENTAL SETTINGS

**Scientific document retrieval.** For each variable, we search the most relevant scientific papers by Google Scholar (SerpApi, 2024), and download up to 20 open accessible papers by the PubMed Central database API (Central, 2024) (see implementation details in Appendix E).

**Baseline methods.** We survey recent LLM-based causal graph recovery methods (see the list in Appendix), and for each dataset, we select the baseline method with the best performance. For each dataset, we present two types of baseline LLMs: baseline LLM1, which is a pure LLM-based method, and baseline LLM2, which is a hybrid method combining a statistical estimation-based and an LLM-based method.

**Validation metrics.** We measure LACR 1 and LACR 2 by different metrics. For LACR 1, we show the the adjacency precision (AP), the adjacency recall (AR), the F1 score, and the Normalized

| | Methods | AP | AP(new) | AR | AR(new) | F1 | F1 (new) | NHD | NHD (new) |
|---|---|---|---|---|---|---|---|---|---|
| ASIA | LACR 1 (BG) | 0.8750 | 1.0000 | 0.8750 | 0.8000 | 0.8750 | 0.8889 | 0.0313 | 0.0313 |
| | LACR 1 (DOC) | 0.7273 | 0.9091 | 1.0000 | 1.0000 | 0.8421 | 0.9524 | 0.0469 | 0.0156 |
| | LACR 1 (CON) | 0.7273 | 0.9091 | 1.0000 | 1.0000 | 0.8421 | 0.9524 | 0.0469 | 0.0156 |
| | Baseline LLM1 | 1.0000 | N/A | 0.8800 | N/A | 0.9300 | N/A | 0.0160 | N/A |
| | Baseline LLM2 | 0.8000 | N/A | 1.0000 | N/A | 0.8900 | N/A | 0.0310 | N/A |
| SACHS | LACR 1 (BG) | 1.0000 | 1.0000 | 0.5000 | 0.6667 | 0.6667 | 0.8000 | 0.0661 | 0.0331 |
| | LACR 1 (DOC) | 1.0000 | 0.7780 | 0.5000 | 0.8750 | 0.6667 | 0.8240 | 0.0661 | 0.0331 |
| | LACR 1 (CON) | 0.6429 | 0.5714 | 0.5625 | 0.6667 | 0.6000 | 0.6154 | 0.0992 | 0.0826 |
| | Baseline LLM1 | N/A | N/A | N/A | N/A | 0.3100 | N/A | 0.6300 | N/A |
| | Baseline LLM2 | 0.5900 | N/A | N/A | N/A | 0.5600 | N/A | 0.1200 | N/A |

Table 1: Performances of LACR 1 under settings: BG, DOC, and CON. We test the performance across both datasets, and compare to baseline methods: ASIA: LLM1: (Jiralerspong et al., 2024), LLM2: (Jiralerspong et al., 2024), SACHS: LLM1: (Zhou et al., 2024), LLM2: (Takayama et al., 2024).

Hamming Distance (NHD), as follows. We define: true positive (TP) as the number of edges correctly recovered; false positive (FP) as the number of edges recovered but not in the ground truth; false negative (FN)as the number of edges in the ground truth but not recovered. Subsequently, we define: AP as $\frac{TP}{TP+FP}$, AR as $\frac{TP}{TP+FN}$, F1 as $\frac{2AP*AR}{AP+AR}$, and NHD as $\frac{FP+FN}{n^2}$, where $n$ is the number of variables. Intuitively, NHD is the number of different edges between two graphs, normalized by $n^2$. In the validation of LACR 2, we simply compute the True Edge Accuracy (TEA), i.e., the ratio of correctly oriented edges among all true positive edges in LACR 1's output skeleton.

**Detailed settings.** We use GPT-4o in our experiments. In the experiments, we evaluate our solution under different knowledge settings for the LLM: (1) BG: using only the LLMs' background knowledge to eliminate the causal existence inconsistency; (2) DOC: using LLMs' background knowledge and the retrieved scientific documents, to eliminate the causal existence inconsistency; (3) CON: using LLMs' background knowledge and the retrieved scientific documents to eliminate both of the causal existence inconsistency and d-separation inconsistency.

## 4.3 EVALUATE LACR 1 AGAINST REFINED GROUND TRUTH

With strong scientific evidence showing the necessity of ground truth update, we adjust the "true" causal graphs used in both datasets, and validate LACR 1 against the modified ground truth causal graphs. Details are presented in Table 1, where metrics with label (new) denote the performance of LACR against the modified ground truth, while the other denotes that against the original one.

**Refinement of the ground truth causal graphs.** We first provide the evidence that suggests updating the ground truth. Especially, we modify the Asia causal graph based on evidence returned by LACR, and modify the Sachs causal graph based on the evidence provided in Sachs et al. (2005).

**ASIA.** We add two causal edges in the Asia causal graph due to the following evidence.
(1) **Smoking v.s. Tuberculosis** In the causal graph recovered in (lau, 1988) (see details in Appendix E), variables Smoking and Tuberculosis are independent since all paths between them are not unblocked due to the existence of colliders. However, LACR returns strong evidence (Horne et al., 2012; Wang et al., 2018; Lindsay et al., 2014; Amere et al., 2018; Quan et al., 2022) showing that these two variables are unblockable, which should be associated.
(2) **Bronchitis v.s. X-ray** In (lau, 1988), Bronchitis and X-ray are indirectly associated via a co-variate Smoking. According to the return of LACR, evidence (Jin et al., 2023a; Ntiamoah et al., 2021; Chen et al., 2020; Nishino et al., 2014; Yazan et al., 2023) shows that X-ray, especially CT scans, can reveal bronchitis, and a large part of the returned documents show the detection of bronchitis of children, especially with the help of deep learning.

**SACHS.** We modify one causal edge in the SACHS causal graph due to the following evidence.
Sachs et al. (2005) evaluates their result against a causal graph provided by biological experts (biological graph). However, their graph is still different from the biological graph, i.e., the causal effect from PKA and PKC to P38 and JNK, respectively, are mediated via exogenous variables.

**Observations against new ground truth.** Table1 presents the performance of LACR 1 across three different knowledge databases (BG, DOC, and CON) for two datasets, ASIA and SACHS. The table highlights the comparison between results based on the original ground truth and the new ground truth, which has been updated according to the latest research findings. Performance is measured using both F1 scores and NHD values, reflecting the revised ground truth. These results are also compared against baseline LLM methods (LLM1, LLM2) for both datasets. From our experiments, we make three key observations:

Firs, across both datasets, the performance of all LACR solutions improves consistently, as seen in both the F1 and F1(new) scores. F1(new) reflects updates to the causal graph based on the latest research results. This shows that LACR can effectively understand and incorporate CARs from related literature, allowing the voting mechanism to contribute to better F1 scores. This demonstrates the strength of our method in extracting and utilizing up-to-date information.

Second, in the ASIA dataset, LACR 1 with BG, DOC, and CON sees improvements of 1.6%, 13.1%, and 13.1%, respectively, when comparing the original and new versions. Notably, DOC, and CON show about 9 times better performance than BG, highlighting the importance of using retrieved knowledge rather than relying solely on the LLM's background knowledge. We observe similar trends on NHD values, with the NHD (new) is never worse than original NHD. Especially, DOC and CON present NHDs (new) around only $1/3$ of the original NHDs, which reinforces the efficacy of LACR. These findings suggest that our solution is more effective than simply injecting new knowledge into LLMs, as the latest SOTA LLM (ChatGPT-4o) is weaker when relying only on background knowledge.

Third, in the SACHS dataset, LACR 1 with BG, DOC, and CON shows significant improvements of 19%, 23.6%, and 2.7%, respectively. Regarding the NHD, BG and DOC achieve the lowest values. The large improvement with BG suggests that it is more reasonable to respect the biological SACHS ground truth, as there is a noticeable gap between the SACHS original dataset and the LLM's background knowledge. On the other hand, the significant improvements with DOC and CON demonstrate that LACR successfully extracts the latest professional knowledge, and the voting mechanism substantially enhances performance by leveraging this updated information.

**Observation against original ground truth.** To fairly compare with the baseline methods, we also evaluate LACR 1 against the original ground truth causal graphs in lau (1988); Sachs et al. (2005). We have the following observations:

**ASIA.** We have three observations from the experimental results on the ASIA dataset. First, both baseline methods slightly outperform LACR 1 regarding the F1 score, with the highest performance achieved by the pure LLM-based method (Jiralerspong et al., 2024). Second, adding retrieved documents into BG reduces performance (AP from 0.8750 to 0.7273, and F1 score from 0.8750 to 0.8421) according to the given ground truth in (lau, 1988), however, it enhances the AR from 0.8750 to 1. Third, by further eliminating the d-separation inconsistency, LACR 1 maintains the performance. Upon checking the LACR's responses, we find that the knowledge of the Asia dataset is considerably rich and clear in the scientific literature and other text corpus, and we conjecture that this is a main reason of pure LLMs' high performance in this dataset.

**SACHS.** We have two observations from the results on the SACHS dataset. First, the best performance of LACR 1 is achieved in settings BG and DOC, outperforming both of the baseline methods, even the hybrid method in (Takayama et al., 2024).Second, eliminating the d-separation inconsistency undermines the performance of LACR 1 (F1 score from 0.6667 to 0.6). The Sachs dataset presents highly professional domain knowledge, with terms easily misunderstood by the LLMs. This is a challenge for the pure LLM-based methods.

## 4.4 LACR 2: ORIENTATION

Table 2 (in Appendix E.4) shows that LACR 2 achieves 1 accuracy in TEA, which indicates 1) it correctly orients all TP edges for both ASIA and SACHS in all settings of BG, DOC, and CON, without need of cycle removal, and 2) the orientation accuracy is consistently high after successfully identifying causal edges with LACR 1, regardless of the knowledge base used. It demonstrates the efficacy of the orientation prompt as well as LLM's capability for causal orientation reasoning. We conjecture that this success is strongly reliant on the rich evidence stored in the scientific literature, which makes the task of orienting edges easier than extracting associational relationships.

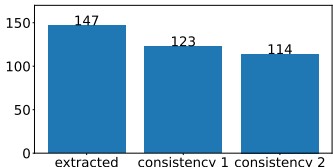 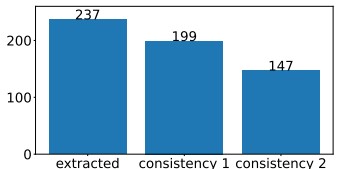

Figure 1: Number of CAR estimation pieces in three phases (Asia: left, Sachs: right).

### 4.5 INCONSISTENCY ESTIMATIONS OF LLMS

We show the inconsistency issue in realistic scenarios by showing the number of extracted CAR estimation pieces used in LACR 1 (Figure 1: "extracted": total CAR estimations extracted by the LLM; (2) "consistency 1": CAR pieces remaining after removing associational inconsistencies; and (3) "consistency 2": CAR pieces remaining after removing both associational and d-separation inconsistencies. Blue and orange bars stand for the Asia and Sachs datasets, respectively.). In the ASIA dataset, out of 147 extracted CAR estimation pieces, 123 (83.7%) passed the associational consistency check (consistency 1), and 114 (77.6%) passed both associational and d-separation consistency checks (consistency 2). For the SACHS dataset, 237 pieces were extracted, with 199 (83.9%) passing consistency 1, but only 147 (62.0%) passing consistency 2. This indicates that SACHS experiences a more significant reduction in adopted CAR estimations after applying the d-separation consistency check compared to ASIA.

Theoretically, applying consistency 1 involves removing minority opinions among the extracted CAR estimation pieces for each pair of factors, thereby reducing causal existence inconsistency. This process ensures that only the majority-supported associations are considered, enhancing the associations' reliability. Consistency 2 checks the validity of indirect associations by examining d-separation sets mentioned in the literature. If inconsistencies are found, such as factors in the d-separation set not being connected to the two factors under investigation, the support for an indirect association is weakened. Consequently, relationships previously considered indirect may be reclassified as direct associations. This shift can lead to an increase in AR, as more associations are identified, but may cause a decrease in AP due to the potential inclusion of false positives.

This theoretical impact is reflected in the performance results shown in Table 1. In the ASIA dataset, both consistency levels are relatively high, with minimal reductions after applying the consistency checks. In contrast, the SACHS dataset exhibits a significant decrease in the number of adopted CAR estimations after applying consistency 2, with 38% of the literature removed due to d-separation inconsistencies. This substantial reduction increases the likelihood of voting for direct associations, as fewer indirect associations are supported by the remaining literature. The increased emphasis on direct associations leads to a rise in FP and a decrease in FN. As a result, the AP decreases from 1.0000 in LACR 1 (DOC) to 0.6429 in LACR 1 (CON), while the AR slightly increases from 0.5000 to 0.5625, as shown in Table 1. This shift reflects the trade-off between precision and recall when inconsistency removal disproportionately affects one type of association over another.

## 5 CONCLUSION

In this paper, we proposed a novel LLM-based causal graph construction method called LACR which uses the constraint-based causal prompt strategy designed according to the constraint-based causal graph construction (CCGC) method. Comparing to most existing LLM-based causal graph construction methods, that use the direct causal prompt to query LLMs to do highly complex causal reasoning, LACR mainly relies on LLMs to do low-complexity associational reasoning, and follows the process of CCGC to determine the causal relationships. For accurate associational reasoning, we retrieve information from external scientific corpus as the context of LLM queries. We evaluate LACR's efficacy on benchmark datasets, particularly,we show LACR is sensitive to the new evidence in the latest literature, which indicates its usefulness for scientific research.

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

# A APPENDIX

## A.1 EXAMPLES

As follows, we first show an example of statistical estimation-based methods' vulnerability to a type of data bias, the so-called selection bias (Bareinboim et al., 2014).

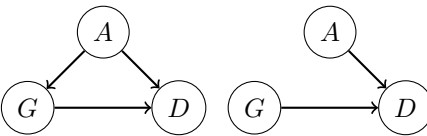

Figure 2: Causal graphs in Example 1: left-the truth causal graph; right-recovered causal graph by the biased data.

**Example 1.** *Consider that we would like to investigate the causal relationship of three variables: $A$ (human age), $G$ (human gender), and $D$ (some disease). Assume that the true causal graph is the left figure in Figure 2.*

*Generally speaking, human age and gender are associated because female has a longer average lifespan. Assuming that this association is only significant for $A \geq 60$. However, if each point in a dataset has age under 60, we cannot observe significant difference between the population of male and female. Then, we would recover the causal graph as the right figure in Figure 2.*

The second example shows the processing of a well-known constraint-based causal graph discovery algorithm called PC algorithm.

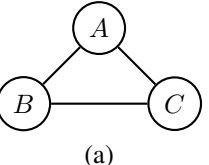

(a)

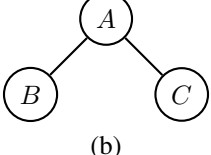

(b)

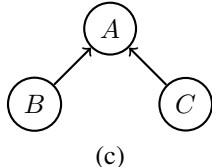

(c)

Figure 3: PC algorithm's process.

**Example 2.** *Consider a causal discovery task for three variables $A$, $B$, and $C$, and two different joint probability distributions $P^1$ and $P^2$. We start with a complete undirected graph Figure (a) 3.*

*Then, by $P^1$, we conduct the zero-order independence tests and obtain: $\hat{\alpha}(AB) = 1$, $\hat{\alpha}(AC) = 1$, and $\hat{\alpha}(BC) = 0$. Then, we keep edges $(A, B)$ and $(A, C)$, and remove $(B, C)$, and obtain Figure (b) 3, since $B$ and $C$ are not a cause of each other, otherwise they must be associated. Based on the zero-order tests, we can already determine the causal graph as Figure (c) 3, as $A$ must be a collider since $B$ and $C$ are d-separated by $\emptyset$.*

*On the other hand, if we consider $P^2$, we first have zero-order tests showing all pairs are associated, and we cannot remove any edge in Figure (a) 3. We then conduct first-order tests, and obtain: $\hat{\alpha}(AB \mid C) = 1$, $\hat{\alpha}(AC \mid B) = 1$, and $\hat{\alpha}(BC \mid A) = 0$. Therefore, we can remove the edge $(B, C)$ from Figure (a) 3, and obatin Figure (b) 3. However, we cannot determine the directions of the edges because all directions of $A \to B \to C$, $A \leftarrow B \leftarrow C$, $A \leftarrow B \to C$ indicate the conditional independences consistent with $P^2$.*

## B  ENHANCING SKELETON ESTIMATION ACCURACY BY LACR

The theory of Wisdom of the Crowd (Grofman et al., 1983) states that if (1) each individual voter can make the correct decision better than random decision (e.g., by a toss), and (2) voters make their decision independently, then, the accuracy of the collective decision made by simple majority monotonically increases with the number of voters. In LACR, each CAR estimation can be seen as a voter. Generally the above conditions tend to be guaranteed because (1) both BG and DOC have high quality and the delivered information is better than random information, and (2) different research papers deliver their results in a relatively independent way because of scientific integrity. Therefore, LACR's decision tends to be more accurate than querying single knowledge base, and it can be improved by adding more relevant documents.

## C  PROOFS

### C.1  PROOF OF LEMMA 1

*Proof.* Let $V'$ be a minimal d-separation set of $v_i$ and $v_j$ in a causal graph $G$. Without loss of generality, we reason that an arbitrary variable $v \in V'$ is associated with $v_i$.

Assume that $v$ and $v_i$ are not associated. Since at least a path between $v$ and $v_i$ exists due to the definition of d-separation, a collider must exist on all paths between $v_i$ and $v$. That is, between $v_i$ abd $v_j$, a collider exists on each path that goes through $v$. Then, if we remove $v$ from $V'$, these paths are still blocked, which contradicts against the assumption that $V'$ is a minimal d-separation set. This completes the proof. $\square$

### C.2  PROOF OF PROPOSITION 1

*Proof.* We first show the mapping from the CAR space of each variable pair $v_i$ and $v_j$ is a surjection. The CAR between $v_i$ and $v_j$ must be one of: (1) independent, i.e., $\alpha(ij) = 0$; (2) associated but there exists a variable set that can d-separate $v_i$ and $v_j$, i.e., $\alpha(ij) = 1$, $\exists V' \subseteq V \setminus \{v_i, v_j\}$, $\alpha(ij \mid V') = 0$; and (3) the association between $v_i$ and $v_j$ is not blockable, i.e., $\alpha(ij) = 1$, $\nexists V' \subseteq V \setminus \{v_i, v_j\}$, $\alpha(ij \mid V') = 0$.

Then, according to Algorithm 1, for the above three cases:

- (1). We append set $\{\texttt{all}\}$ to the d-separation collection.

- (2). We append the corresponding d-separation set $V'$ to the d-separation collection.

- (3). We append set $\{\texttt{none}\}$ to the d-separation collection.

We then show that from the space of the returned d-separation sets by Algorithm 1, the mapping to the space of the existence of the corresponding causal edge $(v_i, v_j)$ is also a surjection.

Apparently, all possible d-separation sets returned by Algorithm 1 can be divided into two types.

- Type 1: $\{\texttt{all}\}$ and $V'$ indicate that the association between $v_i$ and $v_j$ is blockable, and thus there should be no causal edge between the variable pair.

- Type 2: $\{\texttt{none}\}$ indicates that the association between $v_i$ and $v_j$ is not blockable, and thus it suggests there is a causal edge between $v_i$ and $v_j$.

This is also a surjection, and it completes the proof. □

### C.3 PROOF OF LEMMA 2

*Proof.* Since the solution d-separation collection of MAXCON is inconsistency free and it retains a maximal number of CAR estimation pieces, for each variable pair $v_i$ and $v_j$, we only need to check one d-separation set in $S_{ij}$ to determine whether $(v_i, v_j)$ exists in $\bar{E}$. This takes $n(n-1)/2$ times of computation, and the result skeleton is consistent with the result skeleton of the approval voting. □

### C.4 PROOF OF THEOREM 1

*Proof.* We can reduce the Maximum Independent Set (MIS) problem, which is known to be NP-hard, to our problem. Formally, given a graph $G = (V, E)$, the MIS problem is to find the largest subset of vertices $S \subset V$ such that no two vertices in $S$ are adjacent.

**Reduction:** Each vote $\delta_{\text{KB}}(ij)$ in our problem corresponds to a vertex in the graph of the MIS problem. If two votes conflict based on the rules defined, draw an edge between their corresponding vertices. This edge indicates that both votes cannot be adopted simultaneously. Finding the largest set of conflict-free votes in our problem is equivalent to finding the largest independent set in the graph constructed above. Since the Maximum Independent Set problem is NP-hard, and our problem can be reduced to it in polynomial time, our problem is also NP-hard. □

### C.5 BUILDING CONFLICT GRAPH $G$

Here is how to create the conflict graph for Algorithm 2. We first create an empty conflict graph $CG = \langle CS, CE \rangle$. For each $s_i \in CS$, we create a corresponding vertex $v_i$ in $V$. For each $s_i$, we check each $s_j \in CS \backslash \{s\}$ if $s_i$ and $s_j$ have a causal existence inconsistency or a d-separation inconsistency, as defined in Definition 2. If any inconsistencies exist, we create an edge $e_{ij}$ connecting the corresponding vertices $v_i$ and $v_j$. Consequently, we get the conflict graph $CG$.

### C.6 PROOF OF PROPOSITION 2

*Proof.* Let $V$ be the set of vertices in the conflict graph $G$, corresponding to the set of votes $\boldsymbol{\delta}$ in the MAXCON problem. For each vertex $v \in V \backslash S$, it was removed because one of its neighbors $u$ was added to the independent set $S$. Since $u$ has at most $\Delta$ neighbors, we have $|V \backslash S| \leq \Delta|S|$. This implies $|V| \leq (1 + \Delta)|S| \quad \Rightarrow \quad |S| \geq \frac{1}{\Delta+1}|V|$. Since $|\text{OPT}| \leq |V|$, we have, $|S| \geq \frac{1}{\Delta+1}|\text{OPT}|$. This completes the proof. □

# D FULL VERSION LACR 2: ORIENTATION

Based on the skeleton recovered, we query the LLMs to extract the direction of each undirected edge in $\bar{G}$ from the same set of scientific documents for each variable pair. Then, we select a subset of LLMs' extractions to shape a cycle-free directed graph, coinciding with our causal background setting.

## D.1 ORIENTATION KNOWLEDGE EXTRACTION

For each variable pair $v_i, v_j \in V$ such that $(v_i, v_j) \in \bar{E}$, we first use the same background reminder prompt to make the LLMs clarify the meaning of the variables in $V$ with inputting the domain names. Then, we input each retrieved document in $\mathbf{DOC}_{ij}$ as the knowledge context, as well as input a causal direction context that instructs the intuition of causal direction, into the LLMs, and query a simple question "Is $v_i$ a cause of $v_j$, or $v_j$ a cause of $v_i$?" We discard all unusable orientation estimations with an answer of "unknown", and record each usable orientation estimation, i.e., either "$v_i \to v_j$" or "$v_i \leftarrow v_j$", in an orientation collection $D_{ij} \in \mathbf{D}$.

## D.2 ORIENTATION

Apparently, we may also encounter inconsistency issues in the orientation collection $\mathbf{D}$, i.e., the *directional inconsistency* and the *cyclic inconsistency*.

For each variable pair $v_i, v_j \in V$, a directional inconsistency occurs if there are two orientation estimations $d, d' \in D_{ij}$ such that $d$ specifies that $v_i \rightarrow v_j$ and $d'$ specifies that $v_i \leftarrow v_j$. Let $D_{i \leftarrow j}$ and $D_{i \rightarrow j}$ be the subsets of $D_{ij}$ such that each orientation estimation is $v_i \leftarrow v_j$ and $v_i \rightarrow v_j$, respectively, and let $\max(D_{ij})$ be the direction between $v_i$ and $v_j$ that has more number of orientation estimations (ties are broken randomly). A directional inconsistency typically points to the orientation estimations that specify two conflicting directions for a causal edge. A cyclic inconsistency happens if for a set of variables $V' = \{v_1, \cdots, v_k\} \subseteq V$ such that for all $1 \le i \le k$, $(v_i, v_{i+1}) \in \bar{E}$, and an orientation estimation is returned specifying that $v_i \rightarrow v_{i+1}$, where $v_{k+1} = v_1$. That is, a set of orientation estimations shape a directed cycle in the causal graph, which is not permitted under our DAG setting[1].

To avoid directional inconsistency, a straightforward and efficient approach is first to order all edges by weight, then process each edge from the highest to the lowest weight, attempting to orient it based on the orientation estimation. If adding the edge creates a cycle, we reverse its direction, and if a cycle still forms, we remove the edge. This method continues until all edges have been processed. However, it may lead to cascade failures, as incorrectly orienting a high-weight edge early on could impact the orientation of the remaining edges. Clearly, this problem is NP-hard, which can be reduced from the Feedback Arc Set (FAS) problem that aims to find the minimal set of edges whose removal makes a directed graph acyclic, which is analogous to ensuring that the oriented edges in our graph do not form cycles. Therefore, we propose Algorithm 3, an approximation solution, towards selecting a DAG with the maximal orientation estimation subset of $\mathbf{D}$ under the constraints of the directional and cyclic inconsistency. It aims to discard the fewest number of orientation estimations to eliminate both types of inconsistency and outputting a DAG.

The algorithm is initiated by setting the graph $G$ as a complete undirected graph $\bar{G}^c$, an orientation estimation collection $\mathbf{D}$, a d-separation collection $\mathbf{S}$, and setting a weight vector $\mathbf{w}$ as an empty list. Then, from Lines 2-6, we remove each undirected edge such that the corresponding d-separation collection suggests the end node variables can be d-separated by a variable set (including the case $s = \texttt{all}$). For each remained edge $(v_i, v_j)$, we record its weight as the number of d-separation sets in the d-separation collection $S_{ij}$. By Lines 7-12, we orient the undirected edges in $G$ in the order decided by the edges weight (high to low), and the direction of each edge is decided by the dominant orientation estimation in $D_{ij}$, i.e., $\max(D_{ij})$. If orienting an edge results in a directed cycle, we un-orient the edge, otherwise we decide the edge's direction in $G$ and remove its weight from $\mathbf{w}$. From Line 13 to 18, we recheck the remained undirected edges, i.e., those form a directed cycle. From the edge with the highest weight, we first try if we orient it by the reverse direction of $\max(D_{ij})$ still forms a directed cycle. If the orientation does not result in a directed cycle, we decide the edge's direction, otherwise we remove the edge from $G$ and finally return a DAG.

# E ADDITIONAL EXPERIMENT DETAILS

## E.1 LOGIC CONNECTION "EITHER" IN THE ASIA CAUSAL GRAPH

A node "Either" is used in the ASIA causal graph to eliminate the difference of the causal effect of "Tuberculosis" and "Lung Caner" on "X-ray" and "Dysponea". In our implementation, we remove node "Either", and query the variables in the remained set. In the graph construction phase, we add the logic connection, and recover the edges as long as "Tuberculosis" has causal relationship with either of "X-ray" and "Dysponea", and the same process is applied to "Lung Cancer".

## E.2 ADDITIONAL INFORMATION OF BASELINES

Note that most of the good-performing baseline LLM-based methods use GPT-4 in their work, but we use GPT-4o in our experiments. We use this new LLM mainly because it is economic. We tried

---

[1]Our method can be slightly modified if the background causal graph setting is tolerable to directed cycles

---

**Algorithm 3** Consistent orientation

---

1: **Initialization:** $G = \langle V, E \rangle = \bar{G}^c, \mathbf{D}, \mathbf{S}, \mathbf{w} = []$
2: **for** $v_i, v_j \in V$ **do**
3:     **if** $\exists s \in S_{ij}$ s.t. $s = \texttt{none}$ **then**
4:         $\mathbf{w}$.append($w_{ij} = |S_{ij}|$)
5:     **else**
6:         $E = E \setminus \{(v_i, v_j)\}$
7: **while** $|\mathbf{w}| > 0$ **do**
8:     for $v_i$ and $v_j$ s.t. $w_{ij} = \max(\mathbf{w})$, orient $(v_i, v_j)$ as $\max(D_{ij})$ in $G$
9:     **if** no directed cycle in $G$ **then**
10:        $\mathbf{w}$.pop($w_{ij}$)
11:     **else**
12:        un-orient $(v_i, v_j)$ in $G$
13: **while** $|\mathbf{w}| > 0$ **do**
14:     for $v_i$ and $v_j$ s.t. $w_{ij} = \max(\mathbf{w})$, orient $(v_i, v_j)$ as the reverse direction of $\max(D_{ij})$
15:     **if** no directed cycle in $G$ **then**
16:        $\mathbf{w}$.pop($w_{ij}$)
17:     **else**
18:        $E = E \setminus \{(v_i, v_j)\}$ and $\mathbf{w}$.pop($w_{ij}$)
19: **Return:** $G$

---

to use GPT-4 in part of the experiments, and found that GPT-4's performance is never worse than GPT-4o.

We select two baseline methods with the best performances for each of the Asia and Sachs datasets as shown in Table 1, by surveying a series of recent LLM-based causal discovery papers that use at least of Asia and Sachs datasets in the evaluation. Hereby, the papers we survey include the following: Cohrs et al. (2024); Takayama et al. (2024); Vashishtha et al. (2023); Jiralerspong et al. (2024); Zhou et al. (2024). We do not consider the following paper as a baseline method: Khatibi et al. (2024), since we found some of the performances it reports show inconsistent with other existing methods, e.g., LLM's causal discovery performance on Asia dataset is significantly lower than the normal level.

### E.3 DATASET DETAILS

**Scientific document pool construction** In our experiment, we automatically build the pre-retrieved scientific document set for each variable pair (**Initialization** in Algorithm 1) in two steps:

(1) Relevant paper search: We search 40 paper titles by querying "name[$v_i$] and name[$v_j$]" to the Google Scholar engine using the SerpApi (SerpApi, 2024), and rank the papers by the search engine's default relevance ranking.

(2) Paper download: Based on the aforementioned ranked paper title list, we use the PubMed API (Central, 2024) to download the papers. For each paper title, we only download the documents from the PubMed Central (PMC) database (i.e., the open-access database of PubMed). for each variable pair, we download up to 20 documents from the top of the ranked title list (note that some papers are unavailable in PMC).

Causal graphs that are recovered by LACR.

The ground truth causal graphs of all datasets in Section 4.

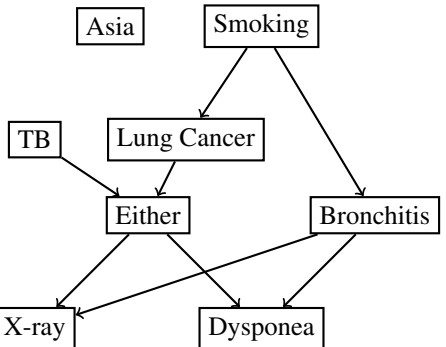

Figure 4: Causal graph of ASIA output by LACR with LLMs' background knowledge.

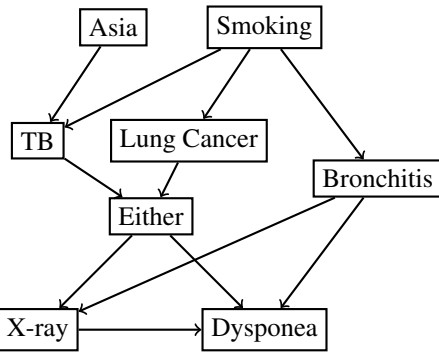

Figure 5: Causal graph of ASIA output by LACR with retrieved scientific documents, and without removal of d-separation inconsistency.

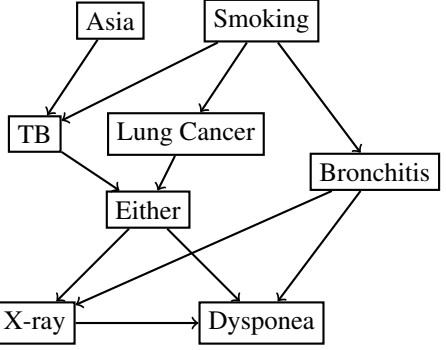

Figure 6: Causal graph of ASIA output by LACR with retrieved scientific documents, and with removal of d-separation inconsistency.

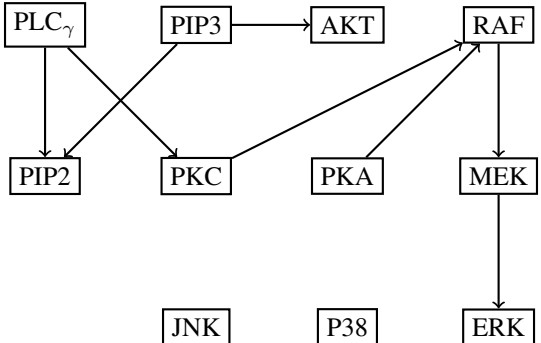

Figure 7: Causal graph SACHS output by LACR with LLMs' background knowledge.

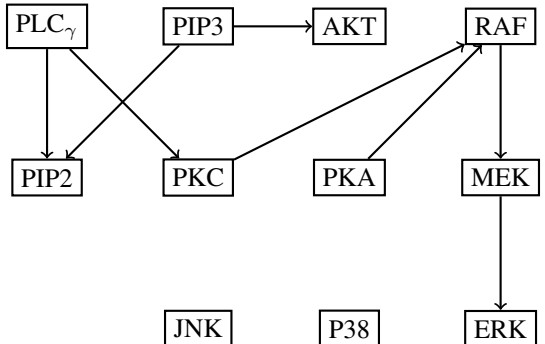

Figure 8: Causal graph SACHS output by LACR with retrieved scientific documents, and without removal of d-separation inconsistency.

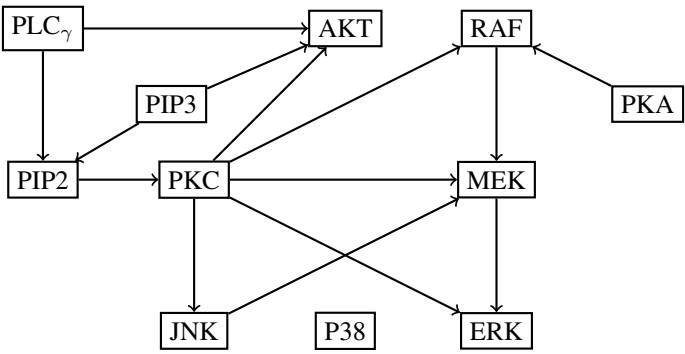

Figure 9: Causal graph SACHS output by LACR with retrieved scientific documents, and with removal of d-separation inconsistency.

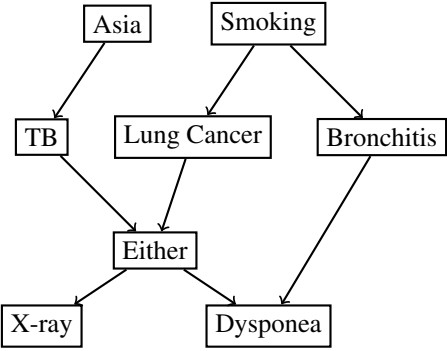

Figure 10: Ground truth causal graph of ASIA in (lau, 1988).

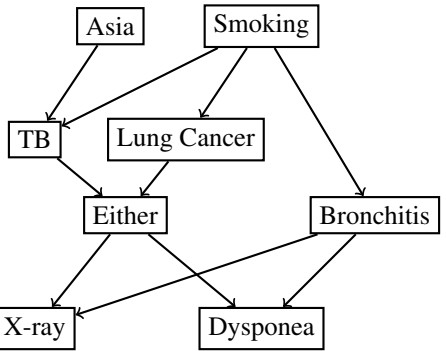

Figure 11: Refined ground truth causal graph of ASIA by LACR.

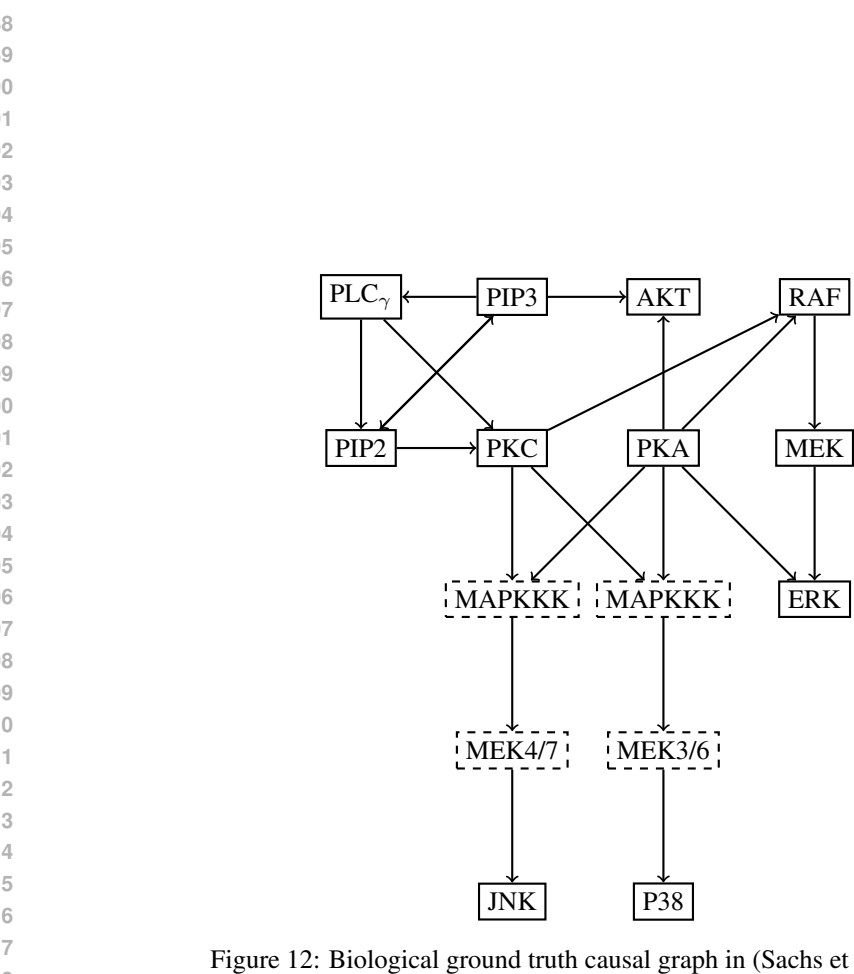

Figure 12: Biological ground truth causal graph in (Sachs et al., 2005).

### E.4 Additional Experimental Results

|            | ASIA | SACHS | ASIA (new) | SACHS (new) |
|------------|------|-------|------------|-------------|
| LACR2 (BG) | 1    | 1     | 1          | 1           |
| LACR2 (DOC)| 1    | 1     | 1          | 1           |
| LACR2 (CON)| 1    | 1     | 1          | 1           |

Table 2: The TEA of LACR 2 on datasets of ASIA, SACHS, based on LACR 1's output skeleton on `BG`, `DOC`, and `CON`, respectively.

### E.5 Additional Experiments for A Modified Asia Dataset.

In this additional experiment, we would like to test the stability of two LLM-based causal discovery methods, namely the simple prompting strategy proposed by Reviewer gpD3, and LACR. We evaluate by using the Asia dataset, which has rich relevant contents in ChatGPT's training data.

Though it seems that ChatGPT has extensive knowledge on the Asia dataset, and it can accurately recover the causal graph by a simple prompt "Give me the edges in the ASIA causal DAG" as mentioned by Reviewer gpD3. We conjecture that ChatGPT cannot reason or extract the accurate relationship between variables aligning with the causal concepts, and instead, it only provides the causal graph by simply repeating the relevant text. To support this conjecture, we slightly modify the variables in the Asia dataset to see whether ChatGPT can recognise the small change of causal relationships. We only change two variables: Visit Asia to Visit US, and X-ray to CT scan. Under this modification, Visit US is not a cause of any other variable in the network because it does not impose change on the risk of obtaining Tuberculosis, however, the other part of the causal graph does not change because X-ray and CT scan can both detect lung diseases. Then, the ground truth causal graph changes to Figure 13. For fair comparison, we run LACR only based on ChatGPT's background knowledge, the same setting for the simple prompt strategy.

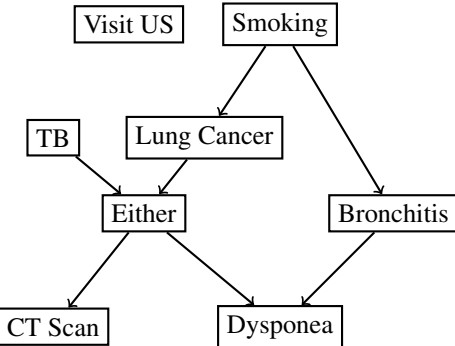

Figure 13: The ground truth used in the additional experiment for modified Asia dataset.

**Simple prompt**   Changes: Visit Asia → Visit US, and X-ray → CT scan.
Query: Show the causal relations for the following variables: Visit to US, Tuberculosis, Smoking, Lung Cancer, Bronchitis, CT scan, Dyspnea.
Answer: "The edges in the causal graph are: Visit to US → Tuberculosis, Tuberculosis → Dyspnea, Tuberculosis → CT scan, Smoking → Lung Cancer, Lung Cancer → CT scan, Smoking → Bronchitis, Bronchitis → Dyspnea.

Visiting the US could reduce the likelihood of tuberculosis compared to areas where tuberculosis is more prevalent. it is identified as causal."

Changes: Visit Asia → Visit US
Query: Show the causal relations for the following variables: Visit to US, Tuberculosis, Smoking, Lung Cancer, Bronchitis, X-ray, Dyspnea.
Answer: "The edges in the causal graph are:
Tuberculosis → Dyspnea, Tuberculosis → X-ray, Smoking → Lung Cancer, Lung Cancer → X-ray, Smoking → Bronchitis, Bronchitis → Dyspnea.

Visit to US → (No direct causal link to tuberculosis or other diseases)."

In this context, visiting the US is unlikely to influence the probability of tuberculosis since the prevalence of tuberculosis in the US is low. However, this variable can be considered irrelevant unless it has a specific causal meaning in this context.

This shows the inconsistency of GPT-4o. The first example demonstrates that GPT-4o remembers ASIA, but does not understand the actual causal relationship. However, LACR is stable on this modification.

**LACR**   Changes: Visit Asia → Visit US, and X-ray → CT scan.
LACR based on ChatGPT's background knowledge returns edges as shown in Figure 14.

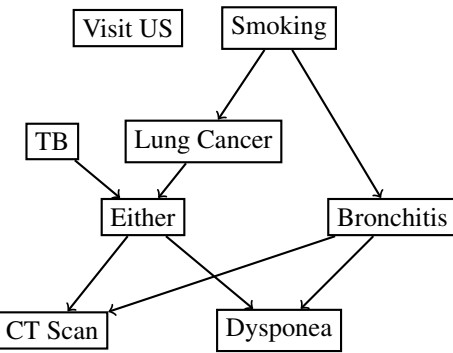

Figure 14: The causal graph returned by LACR based on LLM's background knowledge for the modified Asia dataset.

The detailed metrics for LACR's outputs are shown in Table 3. Notice that these are the performance for the final LACR results. We do not provide the separated validation results for LACR 1 and LACR 2 since we obtain $100\%$ accuracy for LACR 2 (i.e., the orientation phase).

| Methods | AP | AR | F1 | NHD |
|---|---|---|---|---|
| LACR (BG) | 0.8750 | 1.0000 | 0.9333 | 0.0204 |
| LACR (CON) | 0.8750 | 1.0000 | 0.9333 | 0.0204 |
| simple prompt | 1.0000 | 0.8571 | 0.9231 | 0.0204 |

Table 3: The performance of LACR based on LLM's background knowledge.

Observe that LACR can well recognize the slight change of the variables and stably reason the new causal relationships. Notice that an additional edge is recovered by LACR, i.e., Bronchitis - CT scan, validated by this ground truth that is closer to the original ground truth in lau (1988). However, this edge is highly possibly true according to the LACR's responded scientific evidence as shown in Section 4.

### E.6   ADDITIONAL EXPERIMENTS FOR DATASETS ARCTIC ICE COVERAGE AND ALZHEIMER

We additionally conduct experiments for two recent real-world datasets, namely the Arctic Ice Coverage Huang et al. (2021), and Alzheimer Shen et al. (2020).

**Datasets details**   We first describe the two new real-world datasets.
**Arctic Ice Coverage** (Ice): The Ice dataset is introduced in a recent work Huang et al. (2021) from the domains of geography and environmental science, investigating the factors that influence the coverage and thickness of Arctic ice, and the interaction between those factors. The dataset contains 12 variables, which are Sea Ice Coverage and Thickness, Geopotential Height, Relative Humidity, Sea Level Pressure, Meridional Wind At 10m, Zonal Wind At 10m, Sensible Plus Latent Heat Flux, Total Precipitation, Total Cloud Water Path, Total Cloud Cover, Net Shortwave Flux At The Surface, Net Longwave Flux At The Surface. We use the ground truth causal graph identified by Huang et al.

(2021) where the graph contains 39 directed/bidirected edges. Huang et al. (2021) recovers several causal graphs using statistical based causal discovery methods, and we use their results as one of the baseline methods to validate LACR.

**Alzheimer** is introduced in another recent work Shen et al. (2020), in domains of medical science and biology. The work investigates the potential reasons directly or indirectly cause the detection of Alzheimer. The dataset contains 9 variables from four aspects: Demographic variables (Age, Sex, Education Level), biomarkers (Fludeoxyglucose PET, Amyloid Beta, Phosphorylated tau), genetics (APOE epsilon 4 allele), and Diagnosis (Diagnosis of Alzheimer's Dementia). We use the ground truth causal graph identified in Shen et al. (2020) as the ground truth to validate LACR. The ground truth causal graph contains 8 directed edges manually extracted from domain literature. Similarly, Shen et al. (2020) also use several statistical based causal discovery methods to construct the causal graph, and we use their methods as the baseline to compare with.

On both datasets, we run LACR with retrieving maximum of 5 scientific documents for each variable pair.

| | Methods | AP | AR | F1 | NHD |
|---|---|---|---|---|---|
| ICE | LACR (BG) | 0.7368 | 0.4667 | 0.5714 | 0.1458 |
| | LACR (DOC) | 0.6400 | 0.5333 | 0.5818 | 0.1597 |
| | LACR (CON) | 0.6316 | 0.4000 | 0.4898 | 0.1736 |
| | Baseline method | 0.6400 | 0.4103 | 0.5000 | 0.3200 |
| ALZHEIMER | LACR (BG) | 0.5000 | 0.8750 | 0.6364 | 0.0988 |
| | LACR (DOC) | 0.4375 | 0.8750 | 0.5833 | 0.1235 |
| | LACR (CON) | 0.3333 | 0.5000 | 0.4000 | 0.0826 |
| | Baseline method | 0.4600 | 0.6000 | 0.5200 | N/A |

Table 4: Performances of LACR under different settings: BG, DOC, and CON. We test the performance across both datasets, and compare to baseline methods: Ice: the result by DAG-GNN in Huang et al. (2021); Alzheimer: the result by fast greedy equivalence search algorithm in Shen et al. (2020).

**Literature Retrieval Quality Details** We additionally provide details of *usable* retrieved document numbers for variable pairs, to show the sensitivity of LACR on retrieved document quality.

To provide quantified information, we define the *Unknown Ratio* (UR) of all retrieved documents for each variable pair. In LACR, we retrieve a number of scientific documents for each variable pair, and the number is limited by a predefined parameter as described in Section 4.2. However, not all documents can provide useful information to support the CAR decision-making between the variable pair, and LACR returns "Unknown" if the document does not contain relevant contents. Assume that LACR retrieves $k$ documents for a variable pair, and LACR returns "Unknown" for $m$ ($1 \leq m \leq k$) documents. Then, the UR for the variable is $m/k$. The lower is the UR, the more informative documents are retrieved for a variable pair.

Table 5 shows the average UR for three set of variable pairs, namely all variable pairs, true positive variable pairs, and false variable pairs. True positive variable pairs are those that have a causal edge in between in the ground truth causal graph and LACR successfully recovers the edge. A false variable pair denotes that there is a causal edge in the ground truth DAG but LACR fails to recover it, or there is no causal edge in the ground truth DAG but LACR recovers one by mistake. Note that the UR for All variable pairs is not the weighted average of the URs of TP variable pairs and False variable pairs, since we do not report the average UR for true negative variable pairs, i.e., LACR correctly recognizes that no causal edge exists between the variable pair.

| Datasets | Methods | All | TP | False |
|---|---|---|---|---|
| Asia | LACR (DOC) | 0.5058 | 0.3442 | 0.5500 |
| Sachs | LACR (DOC) | 0.4171 | 0.2809 | 0.4714 |
| Ice | LACR (DOC) | 0.8765 | 0.8250 | 0.8478 |
| Alzheimer | LACR (DOC) | 0.9069 | 0.8700 | 0.7917 |

Table 5: The average UR of all variable pairs (ALL), true positive variable pairs (TP), and false variable pairs (False).

**Interpretation** The performance of LACR on the Alzheimer dataset reveals an interesting trend: while AR remains consistent at 0.8750 for both the BG approach and the DOC approach, the AP decreases from 0.5000 (BG) to 0.4375 (DOC). This decline in AP leads to a corresponding drop in the F1 score from 0.6364 (BG) to 0.5833 (DOC), highlighting a negative impact on the overall balance between precision and recall when using retrieved documents. These results indicate that the additional edges introduced in the DOC setting are largely false positives, degrading the quality of the recovered causal graph. Notably, this trend aligns with our earlier observations in other datasets, such as Asia and Sachs, where involving documents initially led to performance drops under outdated ground truth causal graphs.

The Alzheimer dataset's performance behavior supports our hypothesis: when the ground truth graph is outdated and does not reflect the latest scientific consensus, incorporating new knowledge from retrieved documents tends to result in a performance dropping under the old ground truth. As aforementioned, based on trends observed in Asia and Sachs (as discussed in Section 4.5), we notice that, upon updating the ground truth graph to align with the current consensus, the performance in the DOC setting will surpass that of the BG setting. This is because the additional edges introduced by DOC, while penalized under outdated ground truth, are more likely to align with modern causal understandings.

Table 5 offers further evidence for this assumption. While the UR for TP edges is relatively high in Alzheimer's, indicating that BG knowledge dominates the decision to recover most correct causal relationships, the UR for false edges is relatively low, highlighting that retrieved documents are introducing new edges perceived as relevant. This trend is similar to the findings in Asia and Sachs, where incorporating documents initially caused performance drops but, upon aligning the evaluation with updated ground truth, demonstrated the advantage of DOC in leveraging contemporary insights.

Therefore, the observed performance drop for Alzheimer's in the DOC shows our method is sensitive to documents used. This highlights the importance of updating ground truth causal graphs to align with evolving scientific understanding, ensuring a fair and accurate assessment of the added value provided by document-enhanced methods. As seen in other datasets, incorporating up-to-date consensus into the ground truth improves LACR's performance.

It is also worth noting that the overall URs on Ice and Alzheimer datasets are 0.8765 and 0.9069, compared to 0.5058 and 0.4171 for Asia and Sachs datasets. This indicates that the performance drop is the lack of supporting documents. The limited useful documents returned from paper search generate additional noises for the LLM when deciding the causal edges.

### E.7 PROMPTS

### E.7.1 ASSOCIATION CONTEXT

> The association relationship between two factors A and B can be associated or independent, and this association relationship can be clarified by the following principles:
>
> 1. If A and B are statistically associated or correlated, they are associated, otherwise they are independent.
> 2. The association relationship can be strongly clarified if there is statistical evidence supporting it.
> 3. If there is no obvious statistical evidence supporting the association relationship between A and B, it can also be clarified if there is any evidence showing that A and B are likely to be associated or independent statistically.
> 4. If there is no evidence to clarify the association relationship between A and B, then it is unknown.

### E.7.2 ASSOCIATION TYPE CONTEXT

> If two factors A and B are associated, they may be directly associated or indirectly associated with respect to a set of given {third_factors}, and it can be clarified by the following principle:
>
> 1. The first principle is to try to find statistical evidence from the given knowledge to clarify the following association types. If you cannot find statistical evidence, at lease find evidence that is likely to be able to statistically clarify the association type between A and B. If no obvious evidence can be found, the association type is unknown.
> 2. If the evidence shows that A and B are associated via any of the {third_factors}, then A and B are indirectly associated.
> 3. If the evidence shows that by controlling any of the {third_factors}, A and B are not associated any more, then A and B are associated indirectly.
> 4. If the evidence shows that A and B are still associated even if we control any of the {third_factors}, then A and B are directly associated.
> 5. If you think A and B are indirectly associated via any set of the {third_factors}, it must be true that: (1) A and the {third_factors} are associated; (2) B and the {third_factors} are directly associated.
> 6. If you think factors A and B are indirectly associated via other factors, then you must only consider factors in {third_factors}, or at least very similar factors.

### E.7.3 ASSOCIATION BACKGROUND REMINDER

> As a scientific researcher in the domains of {domain}, you need to clarify the statistical relationship between some pairs of factors. You first need to get clear of the meanings of the factors in {factors}, which are from your domains, and clarify the interaction between each pair of those factors.

### E.7.4 LLM ASSOCIATION QUERY (WITH DOCUMENTS)

> Your task is to thoroughly read the given 'Document'. Then, based on the knowledge from the given 'Document', try to find statistical evidence to clarify the association relationship between the pair of 'Main factors' according to the 'Association Context' (delimited by double dollar signs).
> Consider the given document and the association context. Answer the 'Association Question', write your thoughts, and give the reference in the given document. Respond according to the first expected format (delimited by double backticks).
>
> Document:
> {document}
>
> Main factors:
> {factorA} and {factorB}
>
> Association Context:
> $$
> {association_context}
> $$
>
> Association Question:
> Are {factorA} and {factorB} associated?
>
> First Expected Response Format:
> ``
> Document Identifier: XXX

```
Thoughts:
[Write your thoughts on the question]

Answer:
(A) Associated
(B) Independent
(C) Unknown

Reference:
[Skip this if you chose option C above. Otherwise, provide a supporting sentence from the
document for your choice]
``
```

### E.7.5  LLM ASSOCIATION TYPE QUERY (WITH DOCUMENTS)

```
Read and understand the Association Type Context. Consider carefully the role of the {
third_factors} according to the Association Type Context. Based on your thoughts so far,
answer the 'Association Type Question', write your thoughts, and give your reference in the
given document. Respond according to the expected format (delimited by triple backticks)

Association Type Context:
$$$
{association_type_context}
$$$

Association Type Question: Are {factorA} and {factorB} directly associated or indirectly
associated?

Second Expected Response Format:
```
Thoughts:
[Write your thoughts on the question]

Answer:
(D) Directly Associated
(E) Indirectly Associated
(C) Unknown

Reference:
[Skip this if you chose option C above. Otherwise, provide a supporting sentence from the
document for your choice]

Intermediary Factors:
[Skip this if you did not choose D or C above. Otherwise list all factors involved in this
indirect association relationship, each separated by a comma]
```
```

### E.7.6  LLM ASSOCIATION QUERY (WITH BACKGROUND KNOWLEDGE)

```
Your task is to thoroughly use the knowledge in your training data to solve a task. Your task
is: based on your background knowledge, try to find statistical evidence to clarify the
association relationship between the pair of 'Main factors' according to the 'Association
Context' (delimited by double dollar signs).
Consider your background knowledge and the association context. Answer the 'Association
Question', and write your thoughts. Respond according to the 'First Expected Format' (
delimited by double backticks).

Main factors:
{factorA} and {factorB}

Association Context:
$$
{association_context}
$$

Association Question:
Are {factorA} and {factorB} associated?

First Expected Response Format:
``
Thoughts:
[Write your thoughts on the question]

Answer:
(A) Associated
(B) Independent
(C) Unknown
```

```
``
```

### E.7.7 LLM ASSOCIATION TYPE QUERY (WITH BACKGROUND KNOWLEDGE)

```
Read and understand the 'Association Type Context'. Consider carefully the role of any of the
third factors appearing according to the Association Type Context. Then, based on your
thoughts so far, answer the 'Association Type Question', and write your thoughts. Respond
according to the Second Expected Format (delimited by triple backticks)

Association Type Context:
$$$
{association_type_context}
$$$

Association Type Question: Are {factorA} and {factorB} directly associated or indirectly
associated?

Second Expected Response Format:
```
Thoughts:
[Write your thoughts on the question]

Answer:
(D) Directly Associated
(E) Indirectly Associated
(C) Unknown

Intermediary Factors:
[Skip this if you did not choose D or C above. Otherwise list all factors involved in this
indirect association relationship, each separated by a comma]
```
```

### E.7.8 LLM RETHINK QUERY

```
Now, reconsider the association type you answerted above and filter the Intermediary Factors
you found in the following steps:
1. Recheck if your answer aligns with the 'Association Type Context', and if not, revise your
answer.
2. Consider each of the 'Intermediary Factors' you found above. If the factor directly
associates with 'factorA' or 'factorB', then keep the factor in the 'Intermediary Factors'
list, otherwise remove it from the list.
3. Recheck each factor in the refined 'Intermediary Factors' list. If the factor is not in the
'Given Third Factors' list, then remove it from the 'Intermediary Factors' list.
4. Response with the above refined answer, according to the Second Expected Response Format (
delimited by triple backticks).
5. Note that if 'factorA' and 'factorB' are indirectly associated through third factors that
are not in the 'Given Third Factors' list, then the answer is 'Indirect Association', but
return an empty list, that is '[]', for the refined 'Intermediary Factors' list.
Given Third Factors:
{third_factors}

Association Type Question: Are {factorA} and {factorB} directly associated or indirectly
associated?

Second Expected Response Format:
```
Thoughts:
[Write your thoughts on the question]

Answer:
(D) Directly Associated
(E) Indirectly Associated
(C) Unknown

Intermediary Factors:
[Skip this if you did not choose D or C above. Otherwise list all factors involved in this
indirect association relationship, each separated by a comma]
```
```

### E.7.9 LLM DIRECT COVARIATE RETHINK QUERY

```
Now, consider each factors in your returned "Intermediary Factors". According to the "
Association Type Context", consider the following steps and answer the "Direct Intermediary
Factor Question":
1. Recheck the provided document: if it provides any evidence showing that any of the "
Intermediary Factors" directly associated with {factorA} or {factorB}.
```

```
2. If the factor is directly associoated with {factorA}, record it in "Intermediary Factors of
 Factor A.
2. If the factor is directly associoated with {factorB}, record it in "Intermediary Factors of
 Factor B.
4. Response according to the "Final Expected Response Format".

Direct Intermediary Factor Question: Is any factor in the "Intermediary Factors" directly
associated with {factorA} or {factorB}?

Final Expected Response Format:
'''
Thoughts:
[Write your thoughts on the question]

Intermediary Factors of Factor A:
[Return an empty list if no evidence showing any factor directly associated with factorA.
Otherwise list all factors that have a direct association with {factorA} in these square
brakets, each separated by a comma]

Intermediary Factors of Factor B:
[Return an empty list if no evidence showing any factor directly associated with factorB.
Otherwise list all factors that have a direct association with {factorB} in these square
brakets, each separated by a comma]
'''
```

### E.7.10 CAUSAL BACKGROUND REMINDER

```
As a scientific researcher in the domains of {domain}, you need to clarify the statistical
relationship between some pairs of factors. You first need to get clear of the meanings of {
factorA} and {factorB}, which are from your domains, and clarify the interaction between them.
```

### E.7.11 LLM CAUSAL DIRECTION QUERY (WITH BACKGROUND KNOWLEDGE)

```
Your task is to thoroughly use the knowledge in your training data to solve a task. Your task
is: based on your background knowledge, try to find statistical evidence to clarify the
direction of the causal relationship between the pair of 'Main factors' according to the '
Causal direction context' (delimited by double dollar signs).
Consider according to your background knowledge and the 'Causal direction context'. Answer the
 'Causal direction question', and write your thoughts. Respond according to the 'Expected
Format' (delimited by double backticks).

Main factors:
{factorA} and {factorB}

Causal direction context:
$$
{causal_direction_context}
$$

Causal direction question:
Is {factorA} the cause of {factorB}, or {factorB} the cause of {factorA}?

First Expected Response Format:
''
Thoughts:
[Write your thoughts on the question]

Answer:
(A) {factorA} is the cause of {factorB}
(B) {factorB} is the cause of {factorA}
(C) Unknown
'''
```

### E.7.12 LLM CAUSAL DIRECTION QUERY (WITH DOCUMENTS)

```
Your task is to thoroughly read the 'Given document' to solve a task. Your task is: based on
the 'Given document', try to find statistical evidence to clarify the direction of the causal
relationship between the pair of 'Main factors' according to the 'Causal direction context' (
delimited by double dollar signs).
First thoroughly read and understand the Given document and the 'Causal direction context'.
Then, Answer the 'Causal direction question', and write your thoughts. Respond according to
the 'Expected Format' (delimited by double backticks).

Given document:
{document}
```

```
Main factors:
{factorA} and {factorB}

Causal direction context:
$$
{causal_direction_context}
$$

Causal direction question:
Is {factorA} the cause of {factorB}, or {factorB} the cause of {factorA}?

First Expected Response Format:
``
Thoughts:
[Write your thoughts on the question]

Answer:
(A) {factorA} is the cause of {factorB}
(B) {factorB} is the cause of {factorA}
(C) Unknown

Reference:
[Skip this if you chose option C above. Otherwise, provide a supporting sentence from the
document for your choice]
```

