# OpenReview forum: "Consistency Guaranteed Causal Graph Recovery with Large Language Models"
_ICLR.cc/2025/Conference — Submitted to ICLR 2025_

### Official Review · Reviewer_tVED · 2024-10-16

**Soundness:** 2
**Presentation:** 2
**Contribution:** 2
**Rating:** 6
**Confidence:** 3

**Summary:**

This paper aims to recover causal graphs when numerical data is unavailable and individual knowledge is limited.

The main claim: LACR gives better causal graphs than directly instructing LLMs.

The proposed LACR method:
1. infer CARs from documents.
2. recover causal graphs with constraint-based methods.

**Strengths:**

1. An interesting pipeline to construct causal graphs from the scientific corpus.
2. Detailed formalization, analysis, and discussion.

**Weaknesses:**

- The results are not complete. In Table 1, 19 blanks are specified as N/A. I fail to see any difficulty to produce these so-called N/A metrics since F1 score can be produced. Please provide the missing parts.
- The baseline methods used are not consistent. The two baseline methods in ASIA and SACHS are totally different. All four baselines should be fully evaluated in each dataset.
- The evaluation for phase 2 is unclear. In phase 1, two version of ground-truth are used for each metric, like F1 and F1(new). But for phase 2 it is not reported in the same way. Please provide the missing parts.
- The overall evaluation for the final DAG has not been reported. For example, accuracy, recall, F1, SHD, and SID metrics for the 3 variants + 4 baselines over the two used datasets.

**Questions:**

1. There are many important issues in experiments. See the weakness part. These issues make me very worried about the solidness and effectiveness of this paper.
2. Did the authors conduct their own experiment to evaluate the baselines?
3. About motivation. Could you give me any example where scientific papers can be published without supporting numerical datasets? Please clarify the specific scenarios or fields where the method would be most applicable and valuable.


-----

**Post Rebuttal Comments**:

I acknowledge the author's rebuttal.

I have read the others' reviews and rebuttals, I agree that:
- Extracting independent constraints from textual data is interesting and novel. (from reviewer gpD3, 8Gvf)
- The method is useful for integrating the scientific consensus from the retrieved documents and deducing the need for novel datasets. (from reviewer NKix)

I have discussed the evaluation details with the authors.
- Most of the baselines do not have sufficient details to reproduce, as stated by the authors.
- They compared the method with the best-reported numbers in the literature.
- They provided complete causal graphs produced using their method.

This additional information has reasonably alleviated my concerns about the solidity of this paper. Although the current submission still has limitations like "more detailed experimental investigation of the impact of LLMs' capacity" and "reproduced version of baselines," I would like to update my score to 6 and recommend an acceptance.

---

> ### Author Response · Authors · 2024-11-29
>
> We appreciate your comments. Actually, your concern about the incomplete metrics for OUR method does not undermine the soundness of our paper, since the information we provide is equivalent to what you mentioned, i.e., the full metrics for LACR 2 and the final recovered DAGs. We provide the details as follows. For the baseline methods, we cannot report the full metrics due to no accessibility to their codes and result causal graphs.
>
> We response to your comments as follows.
>
> (1) We did not run the code of baseline methods due to no access to the codes and full prompting of existing works. Therefore, we only record the reported metrics. The papers do not provide the constructed causal graphs, and hence we could not have both the precision and recall to compute the F1 score.
>
> (2) We surveyed recent LLM-based causal discovery papers (list in Appendix E.2) and select the two methods with the highest performances. We do not use the same method for both datasets because most of the surveyed papers do not use both datasets for evaluation. By our comparison, we show that our method beats all LLM-based causal discovery methods that we surveyed.
>
> (3) As we described in our experiment results, for both evaluation ground truths and all parameter settings of both datasets, the accuracy of orientation (LACR 2) is $100\%$ which is very stable, and therefore the accuracy, recall, and F1 score are the SAME for the skeleton recovery (LACR 1) and the final DAG. We then chose to report LACR 1's performance in details as this phase mainly determines the overall performance. We rewrote the tables (still, $100\%$ accuracy for all experiments) for LACR 2, for the original ground truth and the refined ground truth, in Appendix.
>
> (4) With numerical data, we can indeed try to use statistical-based causal discovery methods to construct a causal graph, and then conduct consequent causal inference (e.g., ATE estimation). However, this work flow may have crucial problems, e.g, (1) as we mentioned, statistical methods considerably rely on data quality and sufficiency; (2) In a large part of cases, we cannot recovery the DAG without a strong model assumption (e.g., the linear model assumption), leading to non-identification of causality, and therefore we will need external information (e.g., domain expert or literature).
>
> Besides using LACR in the above work flow, our method can significantly benefit other research processing, for instance data collection in empirical research domains, e.g., medical science and social science. To infer causal estimation (e.g., ATE), it is fundamental to estimate the joint distribution of the treatment, the outcome, and all variables in an admissible adjustment set. With a prior causal graph (e.g., given by LACR), researchers can accurately decide the data dimensions for data collection, as it might be considerably expensive to increase data dimensions, or lead to useless data collection once the data dimensions cannot identify the causality of the treatment and the outcome. This is very important for such empirical study domains.

---

> ### Comment · Reviewer_tVED · 2024-12-02
>
> Thanks for the response.
>
> - The authors are right; they provided complete causal graphs produced by their method. (Solved)
> - If I take it right, this paper doesn't reproduce baselines; instead, they pick the best of their reported results. This may have issues with reproducibility. Checking the reproducibility of previous works is definitely a duty of each researcher. Especially: (1) All the baselines have listed their prompts and algorithms in their paper. (2) Many of them are not peer-reviewed.
>
>
>
> It is interesting that skeleton recovery could be a more challenging task than orientation. We know that skeleton and V-structures are identifiable with conditional independence tests. This may suggest that finding independent conditions from literature cannot be easy. It is a good attempt, at least.
>
> I agree that the method has the potential for realistic applications. For example, it can serve as a method for meta-analysis; It can also serve as an alternative when data is not available or very expensive.
>
>
> For the above reasons, I would like to update my score to 5.

---

> > ### Author Response · Authors · 2024-12-04
> >
> > Thank you for your response. As follows, we would like to provide further clarification to your concerns.
> >
> > We select three methods as our baseline, namely, Jiralerspong et al. (2024), Zhou et al. (2024), and Takayama et al. (2024), where most of them only provide the prompt templates. However, the best performance of each paper requires detailed domain knowledge to describe the task and variable meanings, but only one of the papers provides such details. Without the information, the reproduced results are most likely suboptimal compared to the reported results.
> >
> > We found it is indeed more challenging to recover the skeleton than orientation, and existing works (e.g., Kiciman et al. (2023)) also show the high performance in only verifying causal directions. Actually, we think there is rich associational relationship data (including conditional dependency and independency) in the literature, as a large part of the literature does data analysis, such as regression, and such simple analysis is enough for associational relationship verification. Our method retains its reliability and stability by extracting such reliable associational relationship data and doing causal reasoning. In this way, it also has a potential to overcome the bias of individual papers by identifying important latent variables missing in their assumptions to form a causal model, which has further impact other than only solving the issue of data insufficiency.

---

### Official Review · Reviewer_8Gvf · 2024-11-03

**Soundness:** 2
**Presentation:** 3
**Contribution:** 3
**Rating:** 6
**Confidence:** 3

**Summary:**

This paper presents LACR (LLM-Assisted Causal Recovery) for causal graph discovery that leverages LLMs to extract causal relationships from the scientific literature. By combining LLM-driven knowledge retrieval via RAG with constraint-based causal discovery techniques, LACR refines causal graphs with recent literature, addressing data biases and inconsistencies often present in purely statistical methods. Tested on 2 benchmark datasets, the method demonstrates improved causal graph accuracy, showing potential for adaptive, knowledge-rich causal inference.

**Strengths:**

- LLM knowledge might be biased or limited, adding RAG for causal discovery mitigate some weaknesses in the LLMs.

- LACR addresses inconsistency issues in causal relationships using a constraint-based optimization approach, making causal graphs more reliable and less prone to noise from conflicting sources.

- The paper is fairly easy to follow with prompt templates mentioned in the Appendix.

- Novel setup to include RAGs to causal discovery.

**Weaknesses:**

- There is a lack of comparison between statistical (such as PC, FCI, etc) and LLM methods. It would make the paper stronger to have the standard causal discovery evaluations. The paper was motivated against the use of standard methods, it seems like an obvious comparison to make in that case.

- The results have been presented on 2 highly popular datasets - Asia and Sachs. While it is not easy to find datasets that are not ingested by LLM, results on more domains/DAGs would be suggestive of its generalizability.

**Questions:**

- It would be good to mention Limitations and Future Works.

- How would the performance be impacted when lesser capable models are used? Is it still better that standard causal discovery algorithms?

L 367 space needed.


-----
POST REBUTTAL

Apologies for the delay.

I appreciate the authors running PC and other additional experiments. I would like to increase my score to 6. I would have given a higher score if the authors showed the effectiveness of the method with a smaller open source model. However, adding RAG to extract causal relations is still a contribution that will be appreciated by the community. Hence I am increasing the score.

---

> ### Author Response · Authors · 2024-11-29
>
> We thank the reviewer for their valuable comments, and we response to them as follows.
>
> (1) We ran PC algorithm on both of the Asia and Sachs datasets, and obtained a F1 score no more than 0.5 for both datasets. However, we do not include the statistical-based causal discovery methods' performances in our experiment because such comparisons might be not fair. For example, though the causal graph of Aisa dataset was constructed based on real-world data, the accessible data is artificially synthesized based on the graph, NOT the original data. Therefore, the data's distribution aligns with the causal constraints embedded in the original causal graph. However, by our investigation, there exists knowledge gap between the original causal graph and the SOTA domain knowledge, meaning that the current accessible data is biased from the SOTA domain knowledge. Hence, it is unfair to compare with the results developed from such biased data. Other such old dataset may have the same issue, and thus we decide to only compare with LLM-based causal discovery methods.
>
> (2) We mainly chose evaluation datasets from the popular bnlearn causal package. We would like to emphasize that our aim is beyond recover causal graphs that close to the existing "ground truth" which might be out of date. As described in our experiment section (Section 4.3), we observe obvious scientific evidence showing that knowledge gap exists between the original ground truth and the SOTA domain research. LACR shows better capability in bridging such gap, investigating the causal hypothesis development, and updating the ground truth. This is one of the most important contributions of our work.
>
> In this revised version, we include additional experiments for two new datasets (Appendix E.6), namely the Arctic Ice Coverage and Alzheimer. On both datasets, we evaluate against the original statistical based methods, since such new datasets more align with the SOTA knowledge, and it shows that LACR outperforms the baseline methods on both datasets.
>
> (3) One of the main limitations of the current pipeline is the paper retrieval. Please see our new contents in Appendix E.6, where we added the ratio of unusable documents, i.e., the documents cannot provide relevant information. We found that this ratio is high, especially for new causal evaluation datasets. Our next future work is to enhance this by arming LACR with a more efficient information retrieval component. However, we can still see the efficacy of LACR's workflow as it can identify a large part of causal relations correctly even though suffering from lack of scientific documents. We would add an extended version of future works and limitation in the final version.
>
> (4) Since LACR relies on LLM's understanding ability on documents, using a less capable LLM may generate weaker outcomes in respect to evaluating against the baselines. However, our goal is beyond simply constructing a causal graph compared with baselines. One of the most innovative contributions of our approach is its ability to reveal how mainstream causal understandings evolve over time across different periods of literature. By segmenting the corpus into distinct periods (e.g., before 1990, 2000, 2010, or 2020), our method can construct causal graphs that summarize the dominant scientific thinking during each specific period. This provides a dynamic historical view of causal knowledge, allowing us to trace how hypotheses and consensus about key causal relationships have shifted over time, which standard causal discovery algorithms don’t address. For example, in a particular domain, a relationship that was considered causal in earlier periods may be treated as an independent relation in newer analyses or vice versa. The experimental result shown in Table 1 validates this statement, as all the results from our solution improve in the updated dataset (i.e., F1 (new)) compared with the origin dataset (i.e., F1).

---

### Official Review · Reviewer_kqfw · 2024-11-04

**Soundness:** 3
**Presentation:** 2
**Contribution:** 2
**Rating:** 5
**Confidence:** 4

**Summary:**

The paper presents a novel way of developing a causal graph using LLMs, by doing retrieval-augmented generation with scientific documents. They also provide algorithms to resolve inconsistencies in the final causal graph. Experiments are done on two datasets to show the promise of the method.

**Strengths:**

* Good idea to include scientific documents in the prompt of LLM, to avoid solely relying on LLM's background knowledge
* Definition of the two kinds of inconsistencies that appear when merging graphs from different LLM calls
* Algorithmic abstraction of the key consistency challenges faced whenever aggregating inputs from (LLM/human) experts

**Weaknesses:**

* Experiments are done only on two small datasets
* Some choices in the evaluation setup are not well-motivated
* The chosen datasets are unable to show the real potential of the method. Even baselines do well on these datasets. See Table 1 where F1 is the highest for LLM1 baseline for Asia and the F1 is almost the same for LLM2 baseline in Sachs. Instead, it will be good to show experiments on non-memorized datasets (i.e., less popular datasets) where the gains may be higher.

**Questions:**

The formulation of the problem and the algorithmic abstraction are key contributions. I feel that the two algorithms for consistency and orienting direction can be generally useful, even if we are not using any retrieved documents. I have the following questions:

1. The main limitation is that the experiments are not convincing. The choice of datasets is not well-motivated. Both datasets are small graphs and arguably heavily memorized. Choosing another dataset (more complex and less memorized), such as the neuropathic, alzheimers, arctic sea ice, or covid-19 (see kiciman et al. for these datasets) can provide a better motivation (and hopefully stronger results) for the method.
2. Separately, while the main contribution is pitched as the retrieval of documents, I feel that the skeleton building and orienting algorithms are useful in their own right. Often, there are multiple (LLM) experts that may suggest slightly different graphs--would it make sense to do experiments to show that algorithms LACR1 and LACR2 can help any LLM-based method?
3. How many LLM calls are needed to process a variable pair v1, v2? It is not clear from the paper. Is there a sequential process followed. Also, how big is each document? Is a scientific document chunked into paragraphs that is then inserted in the prompt? More details on LLM call time complexity will help. Relatedly, I would be curious to see an ablation where LACR1 and LACR2 are used on top of a baseline LLM algorithm (but without the documents). For example, you can run LLM-BFS with different seeds, or combine LLM-BFS with LLM-pairwise output (assuming that such a combination has similar number of LLM calls as the proposed method). It is difficult to parse whether the gains are due to the documents, or because of LACR1 and LACR2?
4. How are the "best" evaluation baselines decided in Table 1? No justification is provided and the choice seems arbitrary.
5. Why are the "best" baselines not evaluated for the new graphs? This seems unfair. If you are changing the ground-truth based on the output of your own method, at least evaluate the baselines on this new ground truth.
6. Are you assuming causal sufficiency? What if two variables can be d-separated but the separating variable is unobserved. Or if two variables have an unobserved confounder but the algorithm ends up creating an edge between them?

Minor:
There is a typo in the prompt in E.5.1. associtional

---

> ### Author Response · Authors · 2024-11-29
>
> We thank the reviewer for their valuable comments, and we response to them as follows.
>
> (1) We thank the reviewer to provide information of more real-world validation datasets.
> We mainly consider our choice among the networks in the well known bn-learn causal package repository, and select based on two criteria: (1) realistic network because LLM works based on real-world data; (2) large ones among the small scale networks due to the trade-off between running cost/time and evaluation efficacy. For the network memorization issue, (1) we respond Reviewer gpD3's simple prompt strategy by a straightforward modification on the Asia dataset and find that LLM cannot stably identify the apparent causal relations though it seems to have a rich background knowledge (see more details in Appendix E.5 in the revised paper version); (2) LLM's memory may enhance the performance of LACR, however, our focus is definitely beyond recovering causal graphs close to such old ``ground truth'', but specify the causal hypothesis development in literature, and update the graph to fit the SOTA knowledge as we do in our experiments.
>
> Though some of the baseline methods can achieve high performances against the original ground truth, we observe obvious scientific evidence showing that knowledge gap exists between the original ground truth and the SOTA domain research (Section 4.3). LACR shows better capability in bridging such gap, investigating the causal hypothesis development, and updating the ground truth. This is one of the most important contributions of our work.
>
> By your suggestion, we select two new networks: the arctic ice coverage and the alzheimer networks to conduct additional experiments. In the additional experiments, we compare LACR with the statistical methods used in the original papers. Our results show that LACR *outperforms* the best baseline method in both networks with the best F1 scores of 0.5818 and 0.6364 for the Arctic ice coverage and Alzheimer networks, respectively. We added the detailed results in the appendix, in the latest version of the paper. The main limitation of LACR is the effectiveness of scientific document search APIs, where in the current version, a large part of retrieved papers do not contain useful information. We believe that LACR's performance can be significantly improved with a better paper search tool, and this is one of our main future works.
>
> (2) We design these methods for causality. They may have other use-cases, but we didn't consider them at this stage.
>
> (3) We retrieve a number of papers (e.g., at most 20) for each variable pair, and for each paper, we send a sequence of at most 4 queries to LLM as shown in Algorithm 1. We first ask LLM to clarify the meaning of each variable by only giving it the domain name of the task (e.g., medical science), and then we ask LLM to decide whether the variables are associated based only on the paper. If associated, we ask whether this association can be d-separated, and lastly, we ask LLM to recheck and output the d-separation set if the variable pair can be d-separated. Each document is a full scientific paper in pure text downloaded by the PubMed API without chunking, and we limit each document's size by the limitation of the LLM's input limitation. Therefore, the complexity is $4n^2$ times the number of papers for each variable pair. As we do not have access to most of the existing LLM-based causal discovery methods, we add the interesting idea of ablation study of our method against other LLM-based methods in our research agenda.
>
> (4) We surveyed a list of recent LLM-based causal discovery papers (see the list of surveyed papers in our Appendix) that use the same datasets in evaluations, and for each dataset, we select two baseline methods with the highest reported performances. Due to the lack of details (e.g., code or recovered causal graph structure) of the baseline methods, we cannot compute the metrics for the baseline methods against the new causal graphs, and thus we leave the contents N/A.
>
> (5) For latent variables, we do not assume the absence of latent variables as stated in our model (Line 99-100), and in LACR, if any literature identifies exogenous variables that can d-separate a variable pair, we identify the evidence as a support for "no causal edge exists between the variable pair".

---

> > ### Comment · Reviewer_kqfw · 2024-12-03
> > **thanks for the response**
> >
> > Thanks for the response. I am not fully convinced about the memorization issues with bnlearn datasets. I appreciate the experiments on Arctic Sea and Alzheimers datasets. I would suggest the authors to make these the main datasets in a future version of the paper.
> >
> > I think the evaluation still needs to include multiple baseline methods with the same experimental setting, rather than selecting picking 1 or 2 methods. While I agree that some methods are not easy to reproduce, methods like LLM-BFS are available on github and can be implemented easily.

---

> > > ### Author Response · Authors · 2024-12-04
> > >
> > > Thank you for your response. We would like to provide further clarification to your concerns.
> > >
> > > 1. As described in our additional experiment on the Asia dataset (in Appendix E.5), the causal graph produced by an LLM is vulnerable to small prompt changes even though the LLM has a good background knowledge. We have shown that LACR addresses this problem. For example, on Sachs dataset, a simple prompt, such as the one provided by Reviewer gpD3, fails to obtain an accurate result, and adding more domain information cannot improve much, such as our pure LLM-based baseline method. However, LACR, as a pure LLM-based method, can achieve a higher performance than a hybrid method. This indicates LACR can locate the correct information from LLMs' memory without being misled by prompt perturbations.
> > >
> > > 2. As far as we investigated, only LLM-BFS provides code among the methods we surveyed. For almost methods, we can only access the prompt template, however, the detailed input information such the domain specific description of the task and variables is not accessible. We feel it is unfair to reproduce results with incomplete prompts and compare with our method.

---

### Official Review · Reviewer_NKix · 2024-11-04

**Soundness:** 3
**Presentation:** 2
**Contribution:** 2
**Rating:** 5
**Confidence:** 2

**Summary:**

The paper presents a method for recovering causal graphs using LLMs by handling inconsistencies in the LLM's extracted relationships, with this task being formulated as a consistency maximization problem, analyzed theoretically with graph theory tools, and applied on two experimental datasets.

**Strengths:**

There are aspects of the paper I perceive to be strengths.

For example, the LLM prompting strategy seems efficient; the outlined theory is useful in that it helps readers quantify worst-case performance.

The authors explore various evaluation metrics.

The writing is overall clear (although there is some room for possible improvement, see below).

**Weaknesses:**

There are aspects of the paper I perceive to be possible weaknesses, or at least, areas with room for improvement.

The theoretical results seem to be (close to) relatively standard applications of results from graph/approximation theory. Perhaps moer

Some of the algorithms as outlined don't seem to offer much by way of intuition. As with many papers in the DAG-recovery context, there are a number of moving pieces notationally. I would potentially define notation clearly at the head of the algorithms, along with inputs, outputs, and goal. If the associated algorithms run too long, consider moving to Appendix.

One limitation of the evaluation metrics as outlined is that they weigh all edge mispredictions in the same way. In practice, some edge mispredictions in a causal graph may be more or less deleterious in practice. I can think of a few ways this may be overcome in practice. Perhaps the paper selects one relationship in the DAG is of primary scientific interest, and performs ATE estimation with the adjustment set applied by different recovered DAGs. Bias, Variance, and RMSE of the downstream causal estimator(s) could then be examined and could provide useful context for evaluating performance.

Another challenge to contextualizing performance -- I don't seem to see much information for "baseline LLM 1" and "Baseline LLM 2". It is possible, therefore, that the performance gains in Table 1 are due to the specific way of LLM prompting, or the constraint maximization, or the way that majority voting was handled.

Based on the prompting strategy, the "we first retrieve a fixed number of the most relvant scientific papers" seems to be doing a lot of work in the analysis. In general, the proposed method seems to rely on the presence of LLM prior-knowledge of research papers on a given subject. In that sense, the method, as far as understand it, would be difficult to apply in a generic scenario with unlabeled columns. This would imply that the method is much less broadly applicable than competing methods that just use features of statistical distributions of observed variables (I also don't seem to see a comparison with such direct methods). In practice, investigators may have access to papers and so forth; the method described here involves some extra effort to assembled a relevant paper corpus.

**Questions:**

I have some questions about the LLM comparison prompting methods (see above).

**Details Of Ethics Concerns:**

NA.

---

> ### Author Response · Authors · 2024-11-29
>
> We thank the reviewer for their valuable comments, and we response to them as follows.
>
> (1) We appreciate the reviewer’s comment that the theoretical results build on established graph and approximation theory principles. The approximation bound aims to demonstrate that our solution achieves a provable level of performance even under worst-case conditions, which is critical for ensuring robustness and reliability in practice. While we acknowledge that the approximation bound does not aim to be theoretically novel, we believe that this practical perspective highlights the utility of our solution in handling real-world causal discovery challenges.
>
> (2) As we have fully defined and described the notations in the algorithm. We found that it might be more helpful to add an intuitive description to the algorithm. We revised the contents correspondingly and hope it is more clear now.
>
> (3) Indeed, if our purpose is to estimate the causality (e.g., ATE) of a pair of predefined variables (e.g., treatment and outcome), the influence of causal graph estimation's error on the causality estimation highly depends on the causal graph structure. For example, inaccurate edges that are distant from the treatment and outcome in the causal graph may not influence the estimation of causality. However, in this work, our purpose is to construct the causal graph, the purpose of which is not limited to causality estimation, for instance, to guide data collection. Therefore, we treat each misprediction identically.
>
> (4) We surveyed a list of recent LLM-based causal discovery papers (see the list of surveyed papers in our Appendix) that use the same datasets in evaluations, and for each dataset, we select two baseline methods with the highest reported performances. Averagely, other LLM-based methods (including our baseline methods) input much more task specific information in the prompt, however, we do not input such information to keep LACR's generalization ability. Instead, LACR focuses on prompting the statistical intuition to query LLMs, to extract statistical relationships in general tasks. We believe our way of prompting for associational knowledge extraction and the majority voting aggregation can efficiently enhance the performance, but the constraint maximization may benefit or undermine the performance due to information loss as shown in our experiment results.
>
> (5) LACR is indeed sensitive to retrieved scientific documents. However, in our current version, we do not spend extra effort on the paper pool construction, but only use the simple Google Scholar search and PubMed open access paper download APIs for simultaneous document retrieval, as we describe in the paper. Therefore, LACR do not reply LLM's prior knowledge on the retrieved papers, or dimension labels, and it is capable of tackling generic tasks. We additionally show the paper retrieval quality in Appendix E.6 in the revised paper version, and it shows LACR's efficacy even under relatively low retrieval accuracies, especially for the new additional datasets. However, LACR can still outperform most of the existing results. It is worth noting that due to LACR's sensitivity to retrieved papers, we can gain insights into the causal hypothesis development in the task domain, e.g., shown in our modification to the original ground truth graphs. This is one of the most innovative contribution of LACR.

---

> > ### Comment · Reviewer_NKix · 2024-12-02
> > **Thank you for the comments**
> >
> > Thank you to the authors for their responses here. They are helpful. It is true that the main aim is not (necessarily) a contribution to graph theory necessarily. Thanks also for the attempt to clarify the algorithm (although I can't see the revisions).
> >
> > It still might be helpful to run a simulation (at least, for the Appendix) using MAE in ATE estimation as a metric of comparison under some simple assumptions. Usually, not all edges in a DAG are of equal scientific or policy importance, and one edge is of particular interest due to its manipulability. I suspect that the approach here would yield good performance, and which would help convince the non-causal-discovery-focused researchers of LACR's relevance to them.
> >
> > I also tend to agree with Reviewer kqfw's point that "Separately, while the main contribution is pitched as the retrieval of documents, I feel that the skeleton building and orienting algorithms are useful in their own right"; the discussion regarding memorization also was interesting. If the paper were reframed somewhat more clearly as a literature synthesization tool, there would be a less serious concern about memorization (indeed, evaluation is more or less done using the same information also fed to the modeling tool [i.e., scientific literature]). As a synthesization tool, LACR could instead focus on how it integrates and evolves the scientific consensus from the retrieved documents. This reframing might reduce the need for novel datasets while highlighting LACR's robustness in managing conflicting/incomplete information. In this context, memorization would not be a liability but an asset, reflecting the accumulation of historical but evolving knowledge.
> >
> > Reflecting on the paper and on these considerations, I am inclined to maintain my score. In my view, the paper has a contribution; there are also some questions regarding evaluation, memorization, and framing.

---

> ### Author Response · Authors · 2024-12-04
>
> Thank you for your response. We would like to provide further clarification to your concerns as follows.
>
> (1) We added a paragraph to intuitively describe Algorithm 1 from Line 214 to Line 239 in the revised paper version: "Based on the above key prompts, we use Algorithm 1 to extract a CAR estimation piece from each retrieved document if it contains such analyzing result. Intuitively, for each document or LLM’s background knowledge, i.e., KB on Line 3, we query LLM to extract if the KB indicates association or non-association between the variable pair. If the KB indicates association, LLM further investigates whether the association can be blocked or not (Lines 4-6), and we instruct LLM to return the corresponding d-separation set if the association can be blocked (Lines 7-17)."
>
> (2) We appreciate your suggestion of studying the influence of LACR's noise on the ATE estimation. We agree that edges have different importance, and their estimation errors impose different influence on causality estimation (e.g., ATE), based on a predefined scientific scenario. However, the purpose of our work is causal discovery, i.e., constructing causal graphs. Similar to other causal discovery works, e.g., the PC algorithm (Spirtes and Glymour (1991)), the GES algorithm (Chickering (2003)), and the LiNGAM algorithm (Shimizu et al. (2006)), we treat each edge identically. Studying the influence of causal graph estimation error on causality estimation is a very interesting topic, and it is one of our undergoing research projects.
>
> (3) Extracting knowledge from literature for causal graph construction is one of our main contributions, and another is our workflow, i.e., extracting conditional associational relationships (CARs), and conducting constraint based causal discovery reasoning.
>
> In this workflow, LLM memorization is not a issue, and instead, it may enhance the performance of LACR. Reviewer gpD3's concern about LLM's memorization is that they think the performance of LACR is not because of its workflow, but because of LLM's memory of the network. Therefore, it might be possible to simply prompt the LLM to locate the causal graph obtained from the training data. Our additional experiment of manipulating the Asia network (Appendix E.5) shows that such prompting is not reliable. As a comparison, LACR performs stably against the manipulation and recognizes the correct causal relationships between the modified variables.
>
> However, only the combination of accurate document retrieval and LLM's good understanding capability can realize LACR's potential. LACR extracts knowledge from a number of scientific documents as well as from LLM's background knowledge, and aggregates the extracted knowledge to reach a consensus. Based on this feature, the major scientific opinion determines the result. Both of the literature knowledge and LLM's background knowledge are noisy representations of the ground truth, and they together contribute to the rationale of the collective decision making.

---

### Official Review · Reviewer_gpD3 · 2024-11-05

**Soundness:** 1
**Presentation:** 4
**Contribution:** 2
**Rating:** 3
**Confidence:** 4

**Summary:**

The paper introduces LLM Assisted Causal Recovery (LACR), a method for constructing causal graphs using large language models (LLMs) to extract relationships from scientific literature. Building causal graphs from literature is nothing new. The key innovation is a principled constraint-based approach (e.g., using d-seperation) to assemble the DAG from the "Conditional Associational Relationships" extracted from literature to build a causal graph that meets causal discovery principles.

The authors evaluate on the ASIA and SACHS against pure LLM and hybrid methods.

**Strengths:**

The main strength of the paper is the use of causal constraints to evaluate putative causal relationships extracted from text. Prior work has focused mostly on acyclicity constraints, I've never seen a method take constraints to this level of rigor.

**Weaknesses:**

The reliance on faithfulness seems a weakpoint, as faithfulness violations seem more likely when we're just talking about co-occurance in natural language text.

This seems extremely vulnerable to latent variables. There are constraint based approachs that could be used to deal with latents, such as Verma constraints, though good luck getting a natural langauge-based approach to enforcing those constraints.

The evaluation used GPT-4o to reconstruct the SACHS and ASIA networks (also, I believe ASIA was invented as a toy model for causal discovery, which means it is directly discussed in detail in training data). GPT-4o has memorized these DAGs. See below ChatGPT output, I think Sachs is missing a few things (I recall a node called Akt that doesn't appear in my generated text) but GPT's result is close. The ASIA generation is spot on. In my view this invalidates the evaluation:

Here is verbatim from ChatGPT:

USER: * give me a causal DAG with edges from the SACHs network*

ChatGPT: The SACHS network is a causal graph representing protein signaling interactions in immune cells, with edges representing causal influences between proteins. Here’s a simplified version of the SACHS causal DAG structure:

PKC (Protein Kinase C) → RAF
PKC → PIP3 (Phosphatidylinositol (3,4,5)-trisphosphate)
PKC → JNK (c-Jun N-terminal Kinase)
PKC → ERK (Extracellular signal-Regulated Kinase)
PIP2 (Phosphatidylinositol 4,5-bisphosphate) → PIP3
RAF → MEK (Mitogen-activated protein kinase kinase)
MEK → ERK
JNK → c-Jun
ERK → ELK1
This simplified DAG captures some key causal dependencies in the SACHS network. If you need a graphical representation, I can create it with these edges as a visual diagram.

User: *Give me the edges in the ASIA causal DAG*

ChatGPT: The ASIA causal DAG is a well-known example in probabilistic reasoning, often used to illustrate Bayesian networks. It represents the relationships between variables related to a fictional medical diagnosis scenario. Here are the edges in the ASIA causal DAG:

Visit to Asia → Tuberculosis
Smoking → Lung Cancer
Smoking → Bronchitis
Tuberculosis → Tuberculosis or Cancer
Lung Cancer → Tuberculosis or Cancer
Tuberculosis or Cancer → X-ray
Bronchitis → Dyspnea
Tuberculosis or Cancer → Dyspnea
These edges show the dependencies among variables in the ASIA network, illustrating how visiting Asia, smoking, tuberculosis, lung cancer, and bronchitis influence symptoms like dyspnea and the likelihood of a positive X-ray result.

**Questions:**

Comment: Other reviewers might call out the complexity but complexity seems on par with constraint-based discovery algorithms.

In the intro, the authors posit their approach as a solution the bias in the data of causal discovery from numerical data. These seems like an unnecessary contrast, and indeed, natural langauge documents will also have bias (e.g., biases against negative results, biases torwards well-understood systems, etc.). Why not just treat causal graph building from text data as another modality for causal discovery?

LACR optimizes the CARs by removing the minimum number necessary to resolve inconsistencies. Does this induce path dependence in removing CARs?

The fact that Sachs and ASIA DAGs are memorized by GPT-4o is a big problem. Possible remedies:
1. Sachs is a signaling pathway. You can look through biomodels.org or Kegg to find alternative pathways, prompt the model to see if it can reconstruct with high accuracy.
2. Use a smaller opensource model, validate that it hasn't memorized the DAGs, and then use that model.
3. Create an artificial DAG in a science domain, create synthetic corpus based on that DAG.

Willing to upgrade score if this is addressed.

---

> ### Author Response · Authors · 2024-11-29
>
> We thank the reviewer for their valuable comments, and we response to them as follows.
>
> (1) We write faithfulness assumption mainly for the introduction of constraint based causal discovery methods. Obviously, literature based methods can easily violate faithfulness assumption, otherwise there is no inconsistency issue mentioned in our paper. We propose the MaxCon method, i.e., the inconsistency elimination method, aiming at mitigate such violation to some extent.
>
> For latent variables, as we introduced in Section 2.1, we do allow the existence of latent variables as stated in our model. If any literature identifies exogenous variables that can d-separate a variable pair, LACR identifies it as supported evidence for "no causal edge exists between the variable pair".
>
> (2) Our method is fundamentally different from a large part of literature-based causal discovery methods. We would like to stress that LACR does not recover causal graphs directly from individual pieces of literature. Instead, it extracts Conditional Associational Relationships (CARs) for each pair of factors, which is much easier to induce than causal relations, using evidence aggregated from a large corpus of scientific literature. The inferred causal relationships between factors are determined based on broad scientific consensus rather than individual studies. By aggregating knowledge from a diverse set of sources, LACR mitigates the influence of biases or limitations intrinsic to individual papers. We agree that causal graph building from text data can be viewed as another modality for causal discovery. Addressing inconsistency is a key part for fusing this modality, which is exactly our contribution. With different sets of papers published in different time, our method can reveal how causal understandings evolve.
>
> (3) Thanks for raising this interesting point. However, the fact that a dataset is in the training data of the LLM does not mean the LLM knows the causality among the variables. Here are the details regarding your examples:
> (1) The recovered causal graph of SACHS dataset you provided does not match the ground truth, with accuracy of 0.5, recall of 0.25, and F1 score of 0.33. This indicates ChatGPT's weak understanding on such highly professional causal relationships;
> (2) To show that directly prompting ChatGPT cannot provide reliable causal graphs even it uses the data in training, we slightly modify the prompt to observe the vulnerability of causal relations derived from GPT-4o. The prompt recovers a causal graph similar to the ASIA dataset by replacing two variables: Visit Asia $\rightarrow$ Visit US, X-ray $\rightarrow$ CT scan, where Visit US should not be causally related to Tuberculosis. ChatGPT outputs wrong graphs by the simple prompt, specifically connecting Visit US to Tuberculosis. However, LACR can manage such changes and output reasonable causal graphs, where the only ``wrong'' recovered edge (against the original ground truth graph) aligns with the SOTA scientific evidence shown in Section 4.3. The result indicates that simple prompting of ChatGPT does not lead to reliable answers on causality. We show these additional experimental details in Appendix E.5 in the revised paper.
>
> Additionally, we added complementary experiments on two relatively new datasets, namely the Arctic Ice Coverage and Alzheimer, in Appendix E.6 of the revised paper version. The results show LACR outperforms the original statistical based methods in both datasets though the document retrieval quality is low.
>
> (4) Thank you for your insightful question. Resolving inconsistencies in CARs by removing the minimum number of conflicts could be path-dependent if the specific order in which conflicts are resolved affects the overall process. This concern is especially valid in iterative or greedy algorithms, where decisions made early in the process could constrain the options available in subsequent steps.
>
> Despite the potential for path dependence, practically, we observe that the removal process in LACR converges towards consistent results when LACR retrieves a satisfying number of informative documents. Most of the extracted CARs in the dataset reflect a broad scientific consensus, meaning that the number of conflicting CARs is typically small. Hence, the specific choice of which conflicting CARs are removed has only slight impact on the overall structure of the final causal graph. Additionally, the approximation algorithm we use ensures that the retained CARs collectively maximize consistency while satisfying global acyclic and causal constraints. This further mitigates any effects of path dependence, as the removal process is designed to prioritize global consistency rather than being overly influenced by local decisions.

---

### Meta-Review · Area_Chair_9Zjt · 2024-12-21

**Metareview:**

The paper introduces LLM-Assisted Causal Recovery (LACR) to construct causal graphs using LLMs and constraint-based methods.

Strengths:

+ Proposes to use retrieval-augmented generation (RAG) to mitigate biases and knowledge gaps in LLMs for causal discovery.

Weaknesses:

+ Lacks strong empirical evaluation beyond popular datasets, raising concerns about generalizability and potential memorization.

+ Lacks strong comparison with standard statistical causal discovery methods

+ The reliance on LLM prior-knowledge and document retrieval limits applicability in more general or unlabeled scenarios.

**Additional Comments On Reviewer Discussion:**

While the rebuttal addressed some concerns, there remain around lacking detailed enough experimental investigation of the impact of LLMs' capacity and lacking strong enough carisons against other baselines. While some baselines may not have sufficient details to reproduce, it may still be meaningful to compare with a reproduced version of the baselines.

---

### Decision · Program_Chairs · 2025-01-22

Reject